# In vivo imaging of β-cell function reveals glucose-mediated heterogeneity of β-cell functional development

Jia Zhao[1], Weijian Zong[1,2], Yiwen Zhao[1], Dongzhou Gou[1], Shenghui Liang[1], Jiayu Shen[1], Yi Wu[3], Xuan Zheng[1], Runlong Wu[4], Xu Wang[1], Fuzeng Niu[5], Aimin Wang[5], Yunfeng Zhang[4], Jing-Wei Xiong[1], Liangyi Chen[1]*, Yanmei Liu[1,6]*

[1]State Key Laboratory of Membrane Biology, Beijing Key Laboratory of Cardiometabolic Molecular Medicine, Institute of Molecular Medicine, Peking University, Beijing, China; [2]China Department of Cognitive Sciences, Institute of Basic Medical Sciences, Beijing, China; [3]School of Software and Microelectronics, Peking University, Beijing, China; [4]School of Electronics Engineering and Computer Science, Peking University, Beijing, China; [5]State Key Laboratory of Advanced Optical Communication System and Networks, School of Electronics Engineering and Computer Science, Peking University, Beijing, China; [6]Institute for Brain Research and Rehabilitation (IBRR), Guangdong Key Laboratory of Mental Health and Cognitive Science, South China Normal University, Guangzhou, China

*For correspondence:
lychen@pku.edu.cn (LC);
yanmeiliu@pku.edu.cn (YL)

**Competing interests:** The authors declare that no competing interests exist.

**Abstract** How pancreatic β-cells acquire function in vivo is a long-standing mystery due to the lack of technology to visualize β-cell function in living animals. Here, we applied a high-resolution two-photon light-sheet microscope for the first in vivo imaging of $Ca^{2+}$ activity of every β-cell in Tg (ins:Rcamp1.07) zebrafish. We reveal that the heterogeneity of β-cell functional development in vivo occurred as two waves propagating from the islet mantle to the core, coordinated by islet vascularization. Increasing amounts of glucose induced functional acquisition and enhancement of β-cells via activating calcineurin/nuclear factor of activated T-cells (NFAT) signaling. Conserved in mammalians, calcineurin/NFAT prompted high-glucose-stimulated insulin secretion of neonatal mouse islets cultured in vitro. However, the reduction in low-glucose-stimulated insulin secretion was dependent on optimal glucose but independent of calcineurin/NFAT. Thus, combination of optimal glucose and calcineurin activation represents a previously unexplored strategy for promoting functional maturation of stem cell-derived β-like cells in vitro.
DOI: https://doi.org/10.7554/eLife.41540.001

## Introduction

Pancreatic β-cells secrete insulin to regulate glucose metabolism. Insufficient functional β-cell mass leads to glucose intolerance and diabetes. Researchers have intensively studied β-cell development for the last two decades to generate new therapeutic approaches for diabetes. Although many of the mechanisms regulating the early development of pancreatic progenitor cells have been discovered (*Pan and Wright, 2011*), the mechanisms regulating β-cell functional acquisition are still poorly defined (*Kushner et al., 2014*). The β-cells reside in islets containing endocrine, vascular, neuronal and mesenchymal cells. Various signals arising from this neurovascular milieu, such as gap junctions, neuronal transmitters, endothelial factors and hormones, have been reported to be involved in β-cell development (*Borden et al., 2013*; *Carvalho et al., 2010*; *Cleaver and Dor, 2012*; *Omar et al., 2016*). These factors may also directly control β-cell functional acquisition, and the related

**eLife digest** When the amount of sugar in our body rises, specialised cells known as β-cells respond by releasing insulin, a hormone that acts on various organs to keep blood sugar levels within a healthy range. These cells cluster in small 'islets' inside our pancreas. If the number of working β-cells declines, diseases such as diabetes may appear and it becomes difficult to regulate the amount of sugar in our bodies. Understanding how β-cells normally develop and mature in the embryo could help us learn how to make new ones in the laboratory. In particular, researchers are interested in studying how different body signals, such as blood sugar levels, turn immature β-cells into fully productive cells. However, in mammals, the pancreas and its islets are buried deep inside the embryo and they cannot be observed easily.

Here, Zhao et al. circumvented this problem by doing experiments on zebrafish embryos, which are transparent, grow outside their mother's body, and have pancreatic islets that are similar to the ones found in mammals. A three-dimensional microscopy technique was used to watch individual β-cells activity over long periods, which revealed that the cells start being able to produce insulin at different times. The β-cells around the edge of each islet were the first to have access to blood sugar signals: they gained their hormone-producing role earlier than the cells in the core of an islet, which only sensed the information later on.

Zhao et al. then exposed the zebrafish embryos to different amounts of sugar. This showed that there is an optimal concentration of sugar which helps β-cells develop by kick-starting a cascade of events inside the cell. Further experiments confirmed that the same pathway and optimal sugar concentration exist for mammalian islets grown in the laboratory.

These findings may help researchers find better ways of making new β-cells to treat diabetic patients. In the future, using the three-dimensional imaging technique in zebrafish embryos may lead to more discoveries on how the pancreas matures.

DOI: https://doi.org/10.7554/eLife.41540.002

mechanisms need to be studied in vivo. However, the lack of a method to evaluate β-cell function in vivo has hindered the exploration of these fundamental questions. Specifically, β-cell functional development was suggested to be heterogeneous, and this heterogeneous process cannot be studied by simply measuring glucose-stimulated insulin secretion (GSIS) in vivo (*Aguayo-Mazzucato et al., 2011*; *Bader et al., 2016*; *Blum et al., 2012*; *Qiu et al., 2018*; *van der Meulen et al., 2017*). Development of a method to visualize individual β-cell function in vivo will overcome this problem (*Benninger and Hodson, 2018*).

The primary function of a mature β-cell is to quickly secrete stored insulin in response to increases in blood glucose concentrations. Glucose-induced $Ca^{2+}$ influx, which triggers insulin secretion from β-cells, is often used as a functional marker of β-cells (*Pagliuca et al., 2014*; *Rezania et al., 2014*; *Singh et al., 2017*). The extent of $Ca^{2+}$ influx has been used to assess the level of β-cell maturation: the greater the amount of $Ca^{2+}$ influx in, the more mature the β-cells will be (*Rezania et al., 2014*). In isolated islets whose β-cells are labeled with genetically encoded fluorescent $Ca^{2+}$ indicators, $Ca^{2+}$ influx in primary β-cells is imaged as an increase in the fluorescent intensity of the indicators due to their conformational change upon $Ca^{2+}$ binding (*Singh et al., 2017*; *van der Meulen et al., 2017*). However, noninvasively imaging $Ca^{2+}$ transients in β-cells in vivo has not been achieved yet because of the nontransparent pancreas in mammals and the technical challenges in developing high-resolution imaging tools.

Here, we used zebrafish as a model animal because its β-cell development is conserved with mammals in general (*Supplementary file 1*) (*Avolio et al., 2013*; *Qiu et al., 2018*; *Reinert et al., 2014*; *Shah et al., 2011*; *Shih et al., 2013*; *Tiso et al., 2009*; *Yamaoka and Itakura, 1999*) and because β-cell functionality can be observed in its transparent, externally developing embryos (*Huang et al., 2001*). We generated transgenic (Tg) (*ins:Rcamp1.07*) zebrafish, in which every β-cell was labeled with Rcamp1.07 (a red fluorescent $Ca^{2+}$ indicator) and used a homemade high-resolution, two-photon, three-axis, digital scanning light-sheet microscope (2P3A-DSLM) to visualize for the first time the glucose-stimulated $Ca^{2+}$ responses of individual β-cells in vivo. We revealed that a gradually increased glucose concentration, delivered through local diffusion or islet microcirculation,

finely triggered the embryonic β-cells to acquire and enhance their function by activating the calcineurin/nuclear factor of activated T-cells (NFAT) signaling. We further demonstrated that this mechanism was conserved in neonatal mouse β-cell maturation ex vivo, which may promote the functional acquisition and optimal maturity of stem cell-derived β-like cells cultured in vitro. Finally, the variable functions of neonatal mouse islets cultured under different concentrations of glucose speaks for the importance of being able to image β-cell function in vivo, and this technology can be used to study other mechanisms in islet biology, including transdifferentiation, dedifferentiation and regeneration.

## Results

### Visualization of embryonic β-cell function in vivo using 2P3A-DSLM

To visualize β-cell function in vivo, we created a transgenic zebrafish line, Tg (*ins:Rcamp1.07*), in which the red fluorescent calcium indicator Rcamp1.07 (*Ohkura et al., 2012*) is expressed under the control of the insulin promoter. We demonstrated that Rcamp1.07 was exclusively expressed in β-cells, as confirmed by the cellular co-localization of Rcamp1.07 with EGFP in Tg (*ins:Rcamp1.07*);Tg (*ins:EGFP*) double transgenic fish (*Huang et al., 2001*) (*Figure 1—figure supplement 1A–F*) and with immunofluorescently labeled insulin in Tg (*ins:Rcamp1.07*) fish (*Figure 1—figure supplement 1G–J*). As Rcamp1.07 exhibited a very bright basal fluorescence at basal calcium concentrations, all β-cells within the islet were lighted up and readily detectable even in the absence of glucose stimulation (*Figure 1—figure supplement 1C–F*). We then attempted to record the glucose-induced fluorescent intensity change in Rcamp1.07 in a live Tg (*ins:Rcamp1.07*) embryo at 72 hpf, at which stage zebrafish islets have been suggested to regulate glucose (*Jurczyk et al., 2011*). To stimulate β-cells in vivo, glucose was added to the E3 medium to a final concentration of 20 mM to incubate the Tg (*ins:Rcamp1.07*) fish embryos. Within 3 min of stimulation, we observed a robust transient increase in the fluorescence intensity of Rcamp1.07, which indicates glucose-stimulated Ca$^{2+}$ influx in vivo, under a spinning-disc confocal microscope (*Figure 1—figure supplement 2*). However, individual β-cells were difficult to discern under either the confocal microscope or a single-photon selective-plane illuminative microscope (1P-SPIM) (*Figure 1—figure supplement 3* and *Video 1*). Using a two-photon microscope (TPM), we resolved individual β-cells in the XY plane, but the cell boundaries along the Z-axis were blurred because of a low axial resolution and a high scattering of the illumination light in deep tissues (*Figure 1—figure supplement 3* and *Video 1*). Previously, based on 2P-DSLM and the tunable acoustic grin device (*Dean and Fiolka, 2014*; *Duocastella et al., 2012*; *Duocastella et al., 2014*; *Olivier et al., 2009*; *Truong et al., 2011*), we have developed a 2P3A-DSLM (*Zong et al., 2015*). Using this homemade system, we were able to reconstruct a clear three-dimensional (3D) structure of the islet in live fish embryos (*Figure 1A–B* and *Videos 1–3*).

Under 2P3A-DSLM, glucose-responsive β-cells first appeared at 48 hpf in vivo, 24 hr earlier than reported previously (*Jurczyk et al., 2011*; *Singh et al., 2017*). Next, we quantified insulin-positive cells and glucose-responsive β-cells in Tg (*ins:Rcamp1.07*);Tg (*ins:EGFP*) fish embryos at different stages. Interestingly, we identified a progressive increase in glucose-responsive β-cells (from 1.56 ± 0.27 to 11.38 ± 0.49) during the hatching period (48–72 hpf) following an increase in the number of total β-cells (from 14.5 ± 1.45 to 21.88 ± 0.95) during 36–48 hpf (*Figure 1C* and *Figure 1—figure supplement 4A*). We then focused on the hatching period to study the mechanisms by which the embryonic β-cells acquire their function. We used the maximal amplitude (Max ΔF/F0) of glucose-stimulated Ca$^{2+}$ transients that is a key parameter of β-cell function (*Bruin et al., 2015*; *Rezania et al., 2014*), as the criteria to evaluate the functional status of individual β-cells in vivo (*Figure 1—figure supplement 4B*). Compared with the glucose-responsive β-cells appeared at 48 hpf, β-

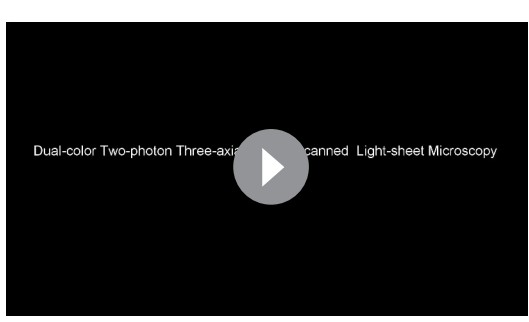

**Video 1.** A schematic illuminating the dual-color 2P3A-DSLM and comparison of image qualities among 1P-SPIM, TPM and 2P3A-DSLM in islet 3D imaging of live zebrafish.
DOI: https://doi.org/10.7554/eLife.41540.008

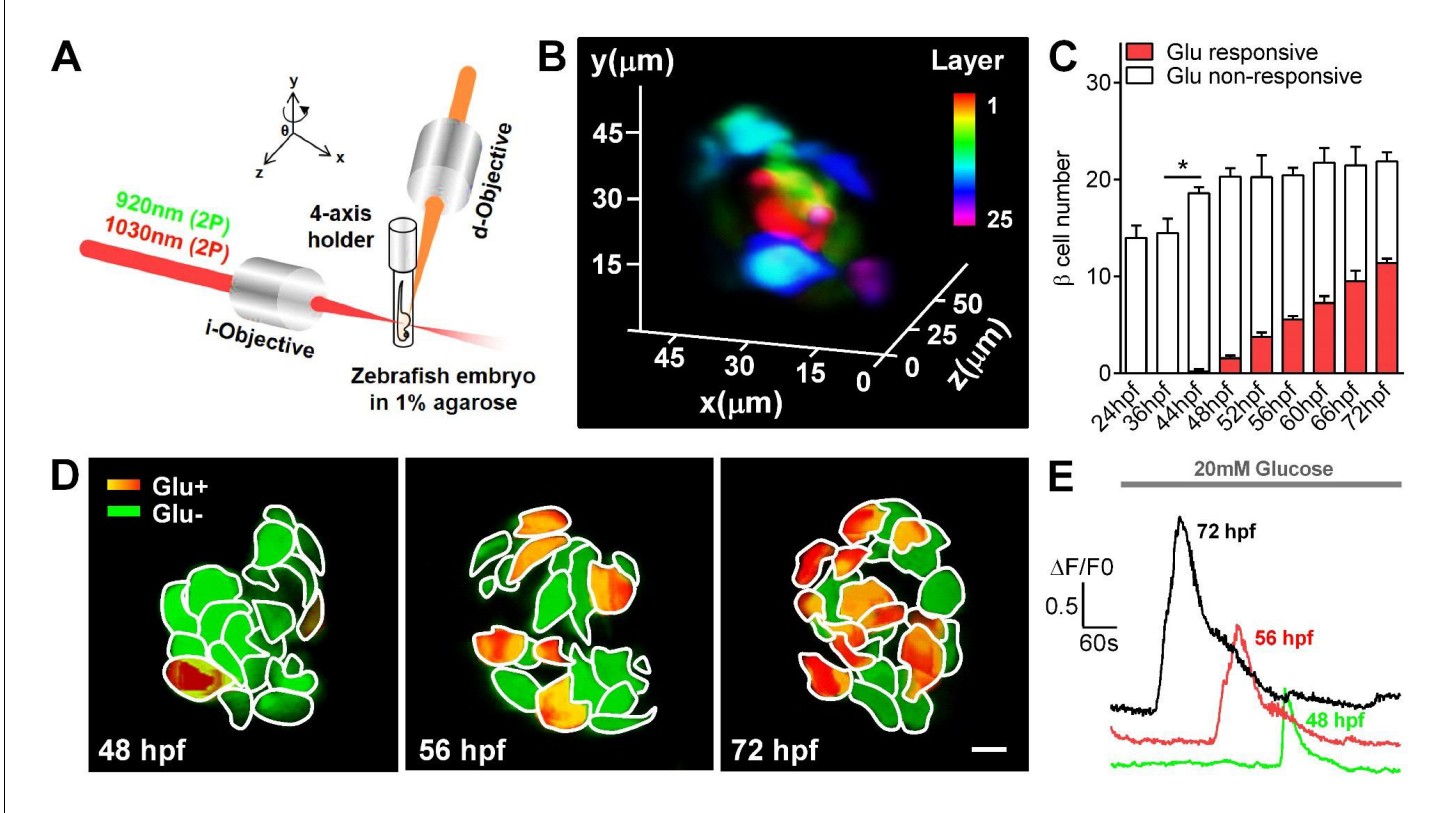

**Figure 1.** Visualization and characterization of β-cell functional development in vivo using 2P3A-DSLM. (**A**) An abbreviated scheme of the dual-color 2P3A-DSLM imaging system. (**B**) 3D-projection of all pancreatic β-cells in a live Tg (*ins:Rcamp1.07*) zebrafish embryo at 72 hpf. The different colors in the color bar represent different depths in the islet. (**C**) Quantification of glucose-responsive and glucose-nonresponsive β-cells at different stages from 24 to 72 hpf in live Tg (*ins:Rcamp1.07*);Tg (*ins:EGFP*) embryos. n = 10–16 embryos per stage. *p<0.05. (**D**) An illustration of the glucose-responsive (red) and glucose-nonresponsive (green) β-cells in islets in live Tg (*ins:Rcamp1.07*) embryos at 48, 56 and 72 hpf. (**E**) Representative traces of glucose-triggered maximum Ca$^{2+}$ transients at the indicated stages. Scale bar: 10 μm. See also *Figure 1—figure supplements 1–4* and *Videos 1–3*.

DOI: https://doi.org/10.7554/eLife.41540.003

The following figure supplements are available for figure 1:

**Figure supplement 1.** Rcamp1.07 was specifically expressed in pancreatic β-cells in Tg (*ins:Rcamp1.07*) zebrafish.

DOI: https://doi.org/10.7554/eLife.41540.004

**Figure supplement 2.** Visualization of glucose-stimulated calcium transients in β-cells in live Tg (*ins:Rcamp1.07*) embryos under a spinning-disc confocal microscope.

DOI: https://doi.org/10.7554/eLife.41540.005

**Figure supplement 3.** Reconstruction of a clear 3D structure of the islet in live zebrafish embryos with our 2P3A-DSLM setup.

DOI: https://doi.org/10.7554/eLife.41540.006

**Figure supplement 4.** Quantification of glucose-responsive β-cells, evaluation of their functional states, and observation of glucose-induced synchronized calcium transients in β-cells in live Tg (*ins:Rcamp1.07*) embryos under 2P3A-DSLM.

DOI: https://doi.org/10.7554/eLife.41540.007

cells at 56 hpf exhibited larger average Ca$^{2+}$ responses (Max ΔF/F0: 93.1% ± 9.7% versus 53.4% ± 8.9%) upon glucose stimulation. These average responses were further enhanced at 72 hpf, at which point β-cells responded with an approximately 150% maximal amplitude of the evoked Ca$^{2+}$ transients (*Figure 1D–E* and *Figure 1—figure supplement 4C*). The histogram of maximal amplitudes of calcium transients from glucose-responsive β-cells shifts right gradually from 48 to 72 hpf, indicating that the increase in calcium signaling of β-cells from 56 to 72 hpf was from the same cells with increased activity (*Figure 1—figure supplement 4C*). Moreover, some neighboring β-cells showed synchronized Ca$^{2+}$ responses in response to glucose (*Figure 1—figure supplement 4D–E* and *Video 2*). Therefore, we witnessed a gradual development of β-cell functionality in vivo from 48 to 72 hpf in zebrafish.

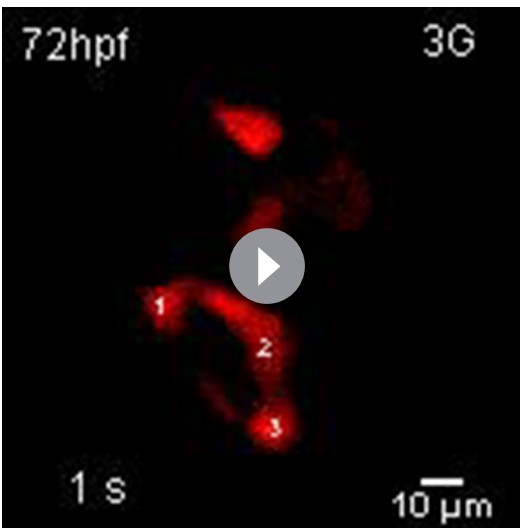

**Video 2.** Representative time-lapse images of glucose-induced synchronized calcium transients in neighboring β-cells in a live Tg (*ins:Rcamp1.07*) embryo at 72 hpf under 2P3A-DSLM.

DOI: https://doi.org/10.7554/eLife.41540.009

## Sequential initiation of β-cell functionality from the islet mantle to the core is coordinated by islet vascularization

Interestingly, by analysing 3D reconstructed images of the Tg (*ins:Rcamp1.07*) zebrafish islets at different stages, we found that β-cells acquired glucose responsiveness sequentially from the mantle to the core of the islet (*Figure 1D*, *Figure 2A* and *Figure 2—figure supplement 1*). From 48–60 hpf, β-cells in the mantle initiated and enhanced their functionality earlier than those in the core and exhibited a higher level of glucose responsiveness (*Figure 2B*) and higher maximum $Ca^{2+}$ transients (*Figure 2C*). From 60 to 72 hpf, β-cells in the core started to initiate and accelerate their functional development. At 72 hpf, β-cells in the mantle and core were indistinguishable in all related parameters (*Figure 2B–C*). These results indicate β-cell functional heterogeneity from islet mantle to the core during the hatching period.

To identify the potential mechanisms underlying the temporal and spatial heterogeneity of β-cell functional development, we visualized β-cells and their adjacent vessels in double transgenic zebrafish Tg (*ins:EGFP*);Tg (*flk1:mCherry*) or Tg (*ins:Rcamp1.07*);Tg (*flk1:GFP*), in both of which β-cells and vascular endothelial cells were labeled (*Wang et al., 2013*). We found that blood vessels initiated contacts with the islet mantle from 48 to 60 hpf and further penetrated into the inner layers of the islet from 60 to 72 hpf (*Figure 2—figure supplement 2A–B* and *Video 4*). Most of the glucose-responsive β-cells were located proximal to blood vessels (*Figure 2—figure supplement 2C–E*), and their longitudinal increase was paralleled by an increase in the number of β-cells located adjacent to the vessels (*Figure 2—figure supplement 2F*). These results indicate that β-cell functional development temporally and spatially correlates with islet vascularization.

To explore whether the heterogeneous development of β-cell function is caused by islet vascularization, we examined β-cell function in the Tg (*ins:Rcamp1.07*); *cloche*[-/-] mutant embryos that have a normal number of β-cells but no vascular endothelial cells or blood cells (*Figure 2D*) (*Field et al., 2003*). At 56 hpf, glucose-responsive β-cells in *cloche*[-/-] embryos were indistinguishable from those in age-matched controls (*Figure 2F*). In contrast, at 72 hpf, *cloche*[-/-] mutants contained fewer glucose-responsive β-cells in the islet core (1.28 ± 0.47 versus 5.51 ± 0.43) and exhibited smaller maximum $Ca^{2+}$ transients in glucose-responsive β-cells (Max ΔF/F0: 59.4% ± 7.8% versus 145.6% ± 8.3%) than the controls (*Figure 2F*). To exclude the possibility that the phenotypes observed above was because β-cells in the islet core did not have access to the glucose stimulation, we incubated embryos with supra-physiological dose of 2-(N-(7-nitrobenz-2-oxa-1,3-diazol-4-yl)Amino)−2-deoxyglucose (2-NBDG, 20 mM) to visualize this fluorescent deoxyglucose analog penetration in the fish embryos. The 2-NBDG (20 mM) efficiently

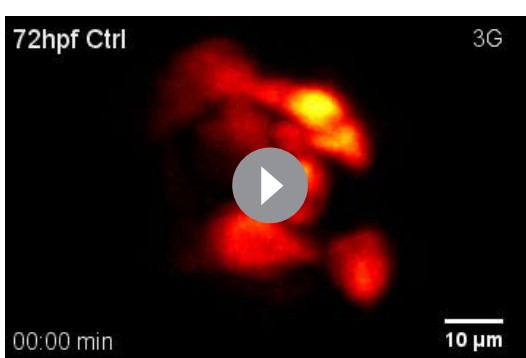

**Video 3.** Representative time-lapse volumetric reconstruction of glucose-stimulated calcium transients in all β-cells in a live Tg (*ins:Rcamp1.07*) embryo at 72 hpf under 2P3A-DSLM.

DOI: https://doi.org/10.7554/eLife.41540.010

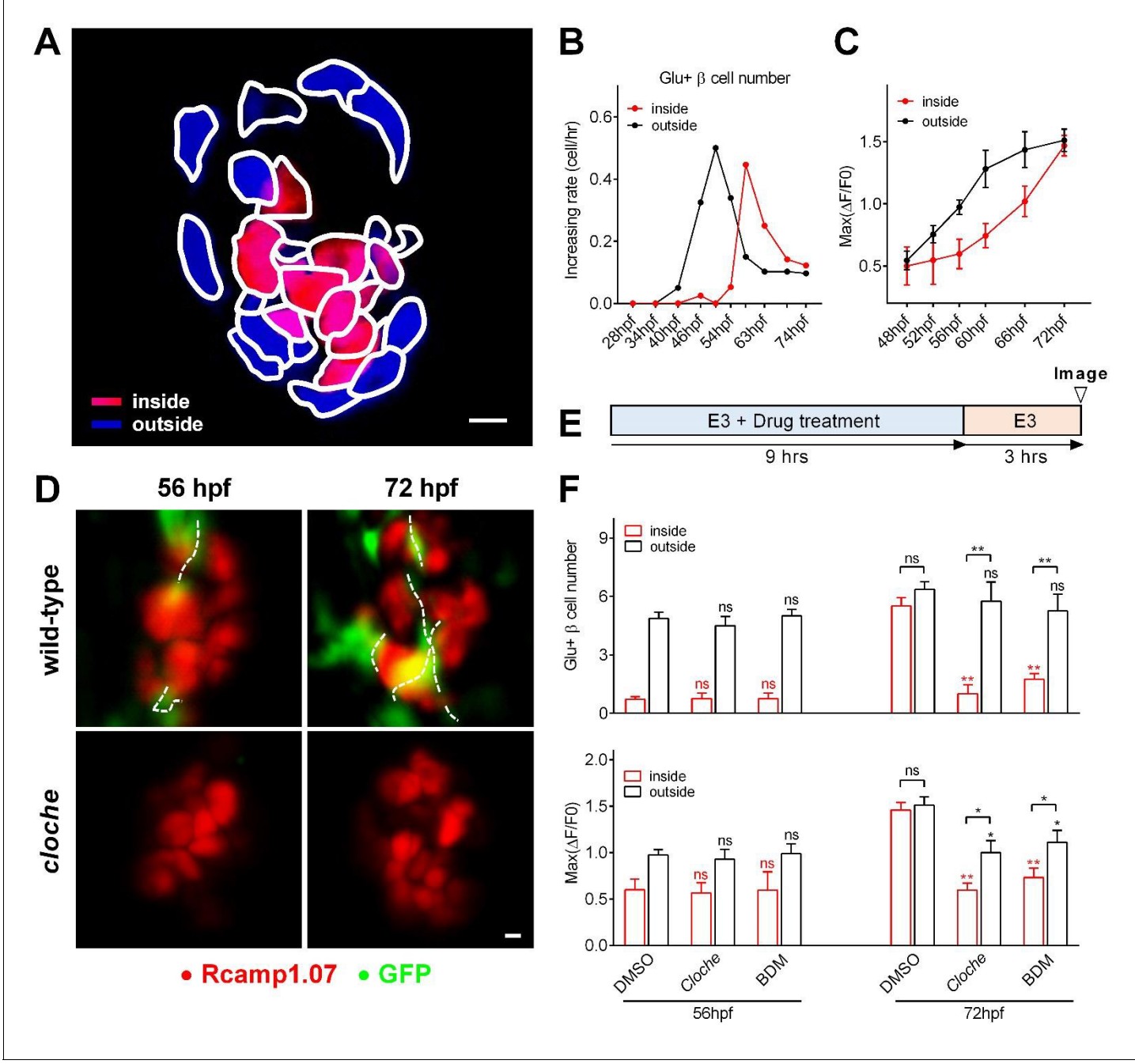

**Figure 2.** Sequential functional acquisition of β-cells from the mantle to the core of the islet was coordinated by islet microcirculation. (A) A schematic illustrating the classification of β-cells into two populations: the cells in the outside (mantle, blue) and the cells in the inside (core, red) of the islet. (B) Time-dependent increases in the number of glucose-responsive β-cells in the mantle (dark) and core (red) of the islet. n = 12–16 embryos per stage. (C) Time-dependent increases in the maximum amplitude of the glucose-triggered $Ca^{2+}$ transients in β-cells from the mantle and the core of the islet. n = 12–16 embryos per stage. (D) Representative 3D-projections of the β-cells (red) and blood vessels (green) in live wild type or *cloche$^{-/-}$* Tg (*ins: Rcamp1.07*);Tg (*flk1:GFP*) embryos. (E) The experimental design for 2,3-BDM treatment used in (F). (F) The number of glucose-responsive β-cells (top) and their maximal $Ca^{2+}$ responses to glucose (bottom) in the mantle and the core of the islets in *cloche$^{-/-}$* mutants and 2,3-BDM-treated embryos. n = 4–6 embryos per stage. *p<0.05, **p<0.01; ns, not significant. Scale bars: 10 μm; scale bars apply to (A) and (D). See also *Figure 2—figure supplements 1–3* and *Video 4*.

DOI: https://doi.org/10.7554/eLife.41540.011

The following figure supplements are available for figure 2:

**Figure supplement 1.** Categorization of β-cells based on their mantle/core localization in the islet.

*Figure 2 continued on next page*

*Figure 2 continued*

DOI: https://doi.org/10.7554/eLife.41540.012

**Figure supplement 2.** Islet vascularization did not affect the total β-cell number but correlated with the sequential acquisition of glucose-responsiveness of β-cells.

DOI: https://doi.org/10.7554/eLife.41540.013

**Figure supplement 3.** Absence of intra-islet circulation caused inefficient delivery of a low concentration of glucose to β-cells in the islet core, whereas either a high concentration of glucose or the complete establishment of islet circulation overcame this deficiency.

DOI: https://doi.org/10.7554/eLife.41540.014

penetrated into the whole islets within 5 min in 56 hpf and 72 hpf *cloche*[-/-] mutants even in the absence of blood circulation (*Figure 2—figure supplement 3A*), indicating that acutely applied high glucose is able to reach all β-cells within the islet independent of islet circulation. Thus, the defective function of β-cells in the islet core of 72 hpf *cloche*[-/-] mutants is due to an arrested maturity of these cells rather than a limited access to high glucose.

Next, we transiently stopped circulation using 2,3-butanedione monoxime (2,3-BDM) (*Bartman et al., 2004*) in wild-type fish for a 9 hr treatment either from 44 to 53 hpf or from 60 to 69 hpf and evaluated β-cell function under 20 mM glucose stimulation at 56 hpf and 72 hpf respectively (*Figure 2E–F*). Although blood circulation was completely recovered during functional evaluation, the blockade of circulation from 60 to 69 hpf significantly impaired β-cell maturity in the islet core (glucose-responsive β-cell number: $1.75 \pm 0.29$ versus $5.51 \pm 0.43$ (control)); Max $\Delta F/F0$: 73.1% $\pm$ 9.9% versus 145.6% $\pm$ 8.3% (control)) to an extent similar to that observed in *cloche*[-/-] mutants at the same age (*Figure 2F*). Therefore, blood circulation, but not the vascular endothelial cells per se, provides a key inductive signal for the initiation and enhancement of β-cell function in the islet core. On the other hand, given that the blockade of circulation from 44 to 53 hpf did not affect β-cells in the islet mantle to acquire glucose responsiveness (*Figure 2F*), blood circulation is not required for the initiation of β-cell functional acquisition in the islet mantle. Nevertheless, we could not exclude the possibility that β-cell functional maturation may cause these cells to secrete factors that promote angiogenesis, or completely eliminate the possible involvement of vascular endothelial cells in β-cell functional development.

## Fine glucose concentrations regulate the heterogeneous development of β-cell function in vivo

Glucose has been reported to regulate embryonic pancreatic endocrine cell differentiation (*Guillemain et al., 2007*). Thus, we investigated whether this major nutrient in the circulatory system also plays a role in the functional development of β-cells. We used 3-mercaptopicolinic acid (3 MPA), an inhibitor of gluconeogenic phosphoenolpyruvate carboxykinase 1 (*pck1*), to suppress endogenous glucose synthesis (*Jurczyk et al., 2011*) for 9 hr from 44 to 53 hpf or from 60 to 69 hpf. This treatment severely inhibited β-cell function in the mantle and the core of the islet under both conditions (*Figure 3* and *Video 5*). Therefore, endogenous glucose is essential for the development of β-cell function in the mantle and the core of the islet. Because yolk syncytial layers, located very close to the islet, already express *pck1* before islet vascularization (*Jurczyk et al., 2011*), locally synthesized glucose may diffuse to the islet mantle to initiate the function of peripheral β-cells in the islet. However, β-cells in the islet core started to acquire function only after the establishment of intra-islet vascularization, indicating that the delivery of inductive concentrations of glucose to β-cells in the islet core may require blood circulation. Indeed, a physiological dose of 2-NBDG (8 mM) did not efficiently reach the islet core at 56 hpf when islet circulation has not been established completely, but fully penetrated into the whole islet at 72 hpf with islet vascularization (*Figure 2—figure supplement*

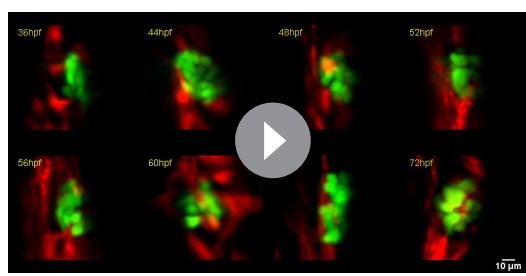

**Video 4.** Volumetric reconstruction of pancreatic β-cells and their nearby blood vessels in live Tg (*ins: EGFP*);Tg (*flk1:mCherry*) embryos from 36 to 72 hpf.
DOI: https://doi.org/10.7554/eLife.41540.015

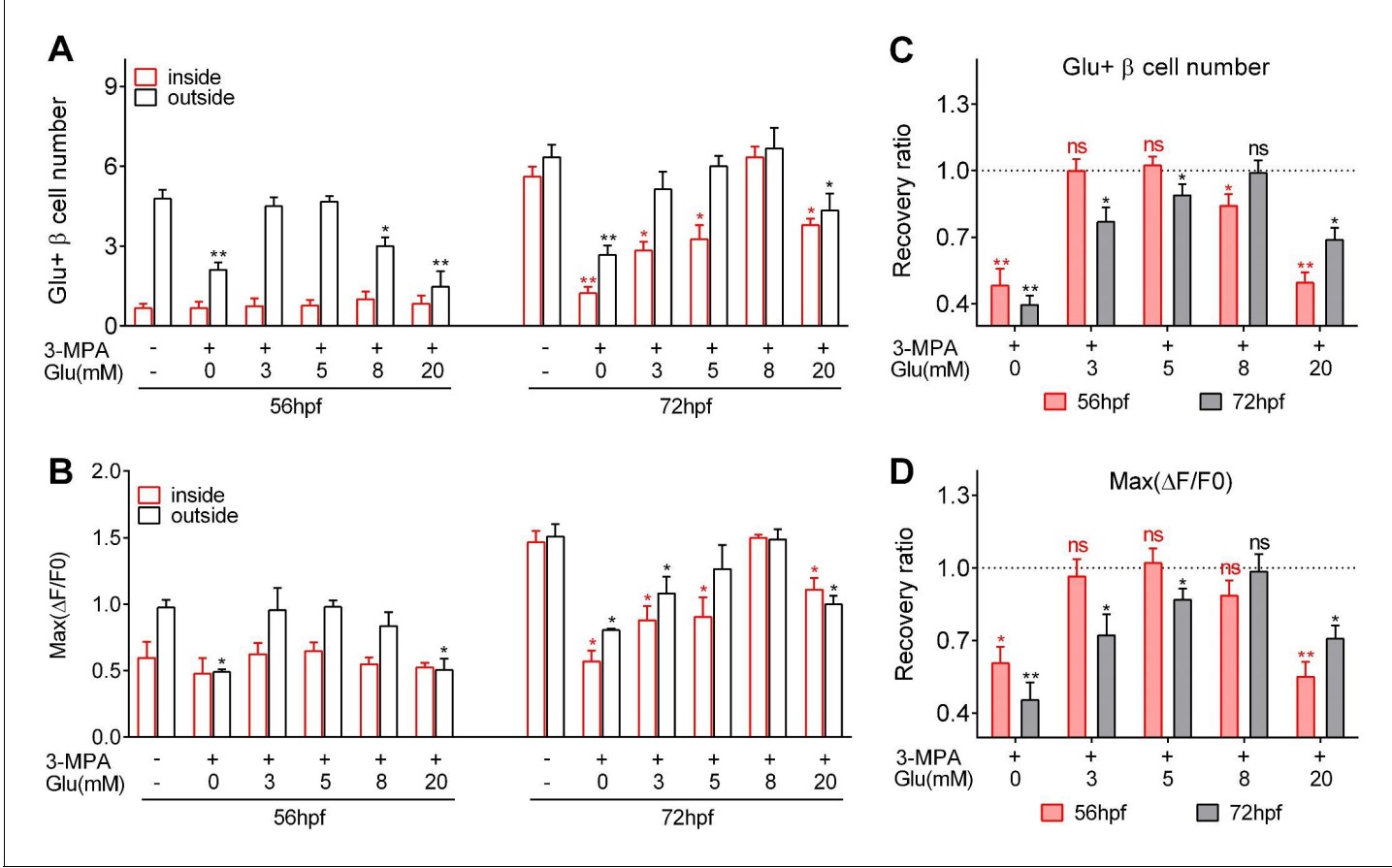

**Figure 3.** Optimal glucose concentrations were required to induce heterogeneous and optimal β-cell functionality at different developmental stages. (A–B) Numbers of glucose-responsive β-cells (A) and their maximal $Ca^{2+}$ responses to glucose (B) in the mantle and the core of the islets from embryos treated with 3 MPA and different concentrations of glucose. n = 4–8 embryos per condition in (A) and (B). *p<0.05, **p<0.01. (C–D) Recovery ratios of glucose-responsive β-cells (C) and their maximal $Ca^{2+}$ responses to glucose (D) following treatment with different concentrations of exogenous glucose in the presence of 3 MPA to inhibit endogenous glucose production. The ratios are presented as the normalized numbers of glucose-responsive β-cells or the normalized maximal amplitudes of the calcium transients compared with those in the control embryos. n = 4–8 embryos per condition in (C) and (D). *p<0.05, **p<0.01; ns, not significant. See also *Video 5*.

DOI: https://doi.org/10.7554/eLife.41540.016

*3B*). These data are in direct contrast to the penetration of supra-physiological dose of 2-NBDG (20 mM) (*Figure 2—figure supplement 3A*) and clearly indicate the role of islet vascularization in delivering a physiological dose of glucose to β-cells within the whole islet.

To explore the optimal glucose concentration for the induction, we incubated zebrafish embryos with a combination of 3 MPA and different concentrations of glucose (3 mM, 5 mM, 8 mM or 20 mM) for 9 hr (*Figure 3*). Treatment with 3 mM or 5 mM glucose from 44 to 53 hpf completely rescued the β-cell functional defects caused by 3 MPA, whereas 8 mM glucose partially restored the number of glucose-responsive β-cells in the islet mantle (*Figure 3*). In contrast, treatment with 8 mM glucose from 60 to 69 hpf completely rescued the phenotype, while 3 mM or 5 mM glucose was insufficient to maintain a normal functional β-cell mass in the islet core (*Figure 3*). In both cases, 20 mM glucose exhibited the worst performance in the rescue experiments, indicating the glucotoxicity of this high concentration under long-term exposure, which has been reported for mammalian β-cells (*Poitout et al., 2006*). However, it is interesting that glucotoxicity in fish embryo caused by chronic incubation of 20 mM is less severe than expected, suggesting that living organism may have some buffering mechanisms to antagonize the glucotoxicity of 20 mM glucose to β-cells. Collectively, glucose triggers the initiation and enhancement of β-cell functionality in both the core and the mantle

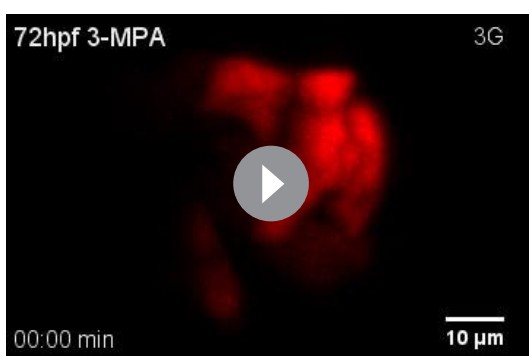

**Video 5.** Representative time-lapse volumetric reconstruction of glucose-stimulated calcium transients in all β-cells in a live Tg (*ins:Rcamp1.07*) embryo at 72 hpf that had been treated with 3 MPA for 9 hr under 2P3A-DSLM.
DOI: https://doi.org/10.7554/eLife.41540.017

of the islet. Moreover, the inductive concentration of glucose is gradually increased with time and mediates the heterogeneous development of β-cell functionality in vivo.

## Glucose-induced calcineurin/NFAT activation to initiate and enhance β-cell functionality

The activation of calcineurin/NFAT by a glucokinase activator was proposed to mediate β-cell development and function in mice (*Goodyer et al., 2012*; *Heit et al., 2006*). Calcineurin is a calcium- and calmodulin-dependent serine/threonine protein phosphatase that activates cytoplasmic NFAT by dephosphorylating it. The activated NFAT is then translocated into the nucleus to regulate downstream gene transcription. To investigate whether calcineurin/NFAT is the signaling pathway downstream of glucose that controls the functional development of β-cells in vivo, we examined β-cell function after preincubating embryos with calcineurin/NFAT inhibitors or activators. Inhibition of calcineurin with FK506 (*Goodyer et al., 2012*; *Heit, 2007*) for 9 hr from 36 to 45 hpf prevented the appearance of glucose-responsive β-cells at 48 hpf without affecting the total β-cell number (*Figure 4A* and *Figure 4—figure supplement 1A*). Inhibition of calcineurin with FK506 or NFAT with VIVIT (*Demozay et al., 2011*) from 44 to 53 hpf or from 60 to 69 hpf significantly reduced the number of glucose-responsive β-cells and their maximal Ca$^{2+}$ transient amplitude at 56 hpf and 72 hpf (*Figure 4B–C* and *Figure 4—figure supplement 1B–C*). On the other hand, activation of calcineurin with CGA or NFAT with ProINDY (*Ogawa et al., 2010*) for 9 hr rescued the defective function of β-cells caused by 3 MPA, 2,3-BDM or the *cloche$^{-/-}$* mutants (*Figure 4B–C*, *Figure 4—figure supplement 1B–C* and *Video 6*). These results suggest that calcineurin/NFAT mediates the role of glucose on triggering β-cell functional development.

To exclude the possible off-target effects of the pharmacological reagents, we did a control experiment by testing the effects of the pharmacological reagents (10 mM BDM, 3 mM 3 MPA, 10 μM FK506 and 141.2 μM CGA) on calcium activities of neurons in the central nervous system (CNS) using Tg (*elavl3:Gcamp6s*) zebrafish in which neurons are labeled by the calcium indicator Gcamp6s (*Dunn et al., 2016*). The result showed that the calcium levels of the CNS neurons were not affected by any of the pharmacological reagents, indicating that the pharmacological treatment in our study specifically affected β-cells (*Figure 4—figure supplement 2*). In addition, we designed a dominant-negative *zebrafish calcineurin A* (*dn-zCnA*) using a similar strategy described previously (*Faure et al., 2007*) and generated transient Tg (*ins:EGFP-GSG-T2A-dn-zCnA*) zebrafish embryos on the Tg (*ins:Rcamp1.07*) genetic background. In these embryos, *dn-zCnA* is co-expressed with EGFP through a *GSG-T2A* linker under the control of the insulin promoter. In total, 75.8% ± 2.5% of Rcamp1.07-positive cells co-expressed EGFP, indicating that 70%–80% of β-cells expressed *dn-zCnA* in Tg (*ins:Rcamp1.07*); Tg (*ins:EGFP-GSG-T2A-dnCnA*) double transgenic zebrafish (*Figure 4D*). However, in these embryos glucose-responsive β-cells appearing at 48 hpf were all EGFP negative, and their number was similar with that in the age-matched controls

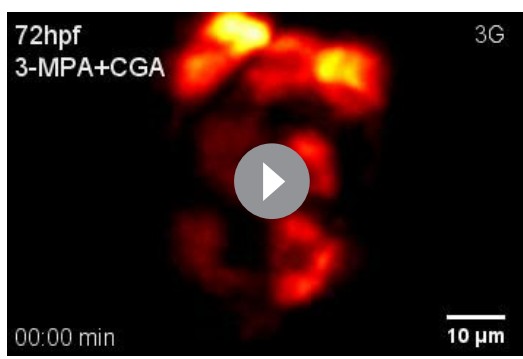

**Video 6.** Representative time-lapse volumetric reconstruction of glucose-stimulated calcium transients in all β-cells in a live Tg (*ins:Rcamp1.07*) embryo at 72 hpf that had been co-treated with 3 MPA and CGA for 9 hr under 2P3A-DSLM.
DOI: https://doi.org/10.7554/eLife.41540.021

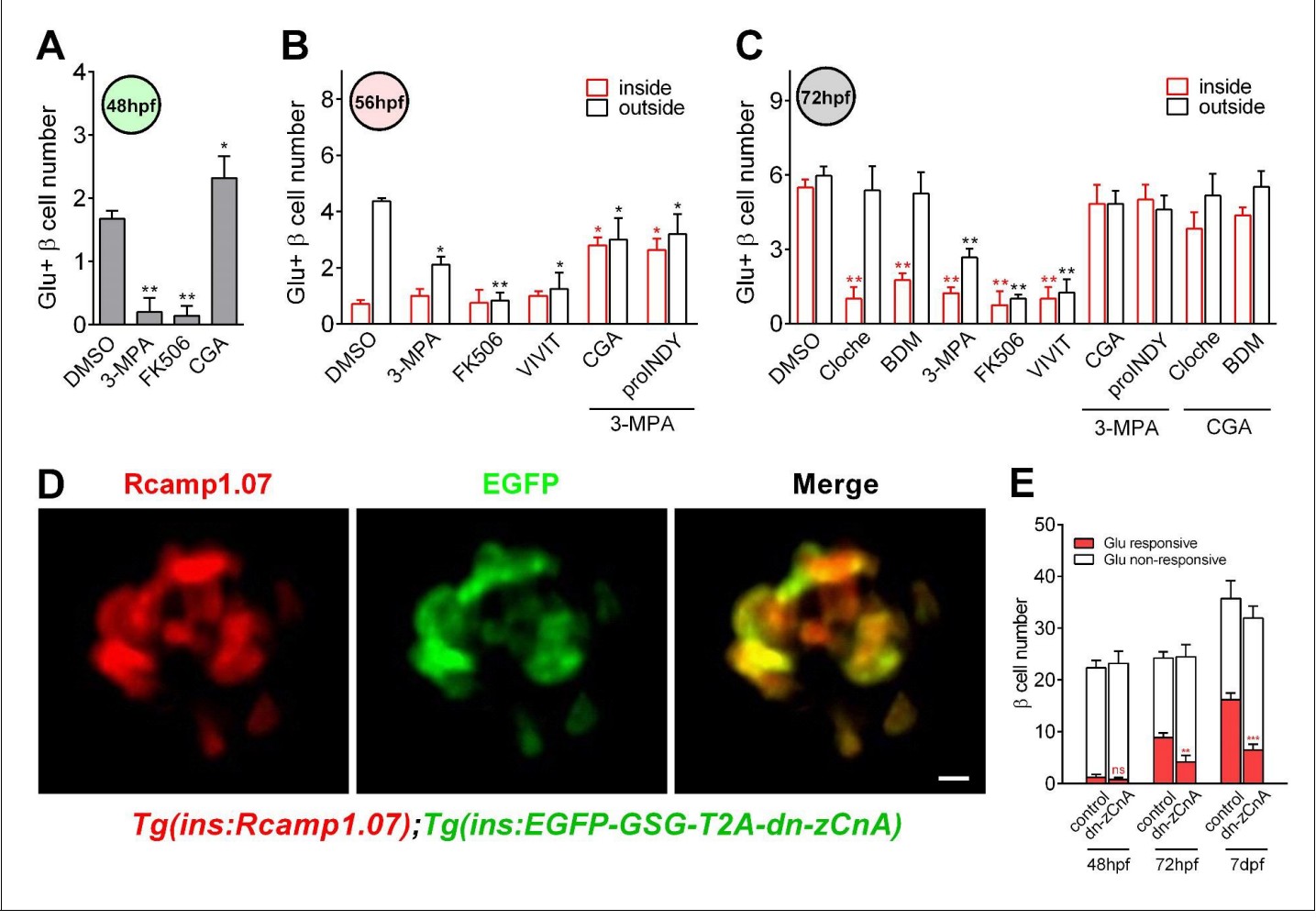

**Figure 4.** Calcineurin/NFAT signaling acted as the downstream of glucose to initiate and enhance β-cell functionality. (A) Numbers of glucose-responsive β-cells in 48 hpf embryos that had been treated with the indicated reagents. n = 4–8 embryos per condition. *p<0.05, **p<0.01. (B) Numbers of glucose-responsive β-cells in the mantle and the core of the islet in 56 hpf embryos that had been treated with the indicated reagents. n = 5–9 embryos per condition. *p<0.05, **p<0.01. (C) Numbers of glucose-responsive β-cells in the mantle and the core of the islet in 72 hpf embryos that had been treated with the indicated reagents. n = 5–9 embryos per condition. **p<0.01. (D) Representative two-photon images of Rcamp1.07 (left), EGFP (middle) and the merged image (right) of the islet cells in a living Tg (*ins:Rcamp1.07*);Tg (*ins:EGFP-GSG-T2A-dn-zCnA*) embryo. (E) Numbers of glucose-responsive β-cells in age-matched controls and *dn-zCnA*-expressing embryos at 48 hpf, 72 hpf and seven dpf. n = 4–6 embryos per stage. **p<0.01, ***p<0.001; ns, not significant. Scale bar: 10 µm. See also *Figure 4—figure supplements 1–2* and *Videos 5–6*.
DOI: https://doi.org/10.7554/eLife.41540.018

The following figure supplements are available for figure 4:

**Figure supplement 1.** Glucose activated calcineurin/NFAT to enhance and sustain the functionality of β-cells during the late hatching period.
DOI: https://doi.org/10.7554/eLife.41540.019

**Figure supplement 2.** Pharmacological treatments did not affect calcium activities of CNS neurons in living Tg (*elavl3:Gcamp6s*) zebrafish embryos.
DOI: https://doi.org/10.7554/eLife.41540.020

(*Figure 4E*). In contrast, the number of glucose-responsive β-cells (70%–80% expressing *dn-zCnA*) was reduced (4.13 ± 1.28 versus 8.87 ± 0.88 (control)) and the maximum $Ca^{2+}$ transients were lower in embryos expressing *dn-zCnA* at 72 hpf than in age-matched controls (78.7% ± 6.9% versus 112.3% ± 10.3%) (*Figure 4E*), suggesting that genetic perturbation of calcineurin/NFAT signaling prevents the glucose-responsiveness of β-cells. To rule out the possibility of the non-specific effects slowing the appearance of markers in the cells, we examined the numbers of glucose-responsive β-cells and their maximal calcium responses to glucose in late (seven dpf) embryos. From 72 to 7 dpf, the glucose-responsiveness of β-cells in *dn-zCnA* zebrafish did not catch up but were still much less

than that in 72 hpf control (*Figure 4E*). Therefore, all of these results demonstrate that calcineurin/NFAT signaling, downstream of glucose, is critical for β-cell functional development.

### Direct activation of calcineurin promotes the high-glucose-stimulated secretion but does not reduce the low-glucose-stimulated secretion of neonatal β-cells in isolated mouse islets in vitro

To explore the effect of calcineurin/NFAT signaling in β-cell functional development in mammals, we isolated neonatal islets from postnatal day 0 (P0) mice and cultured them for 3 days in media supplemented with different concentrations of glucose in the presence or absence of CGA. Then, we measured insulin secretion sequentially in response to either low (3 mM) or high (20 mM) glucose stimulation. When islets had been cultured in medium with 5.6 mM or 7 mM glucose, in the presence or absence of CGA, neonatal islets secreted significantly more insulin than adult islets under 3 mM glucose stimulation (*Figure 5A*), which agreed with previous results that the neonatal β-cells are more sensitive to low glucose (*Blum et al., 2012*). For neonatal β-cells cultured in medium with 11 mM glucose, their insulin secretion at low glucose dropped to a level similar to that of adult β-cells (*Figure 5A*). This finding indicates that optimal glucose may trigger β-cell maturation by reducing their sensitivity to low glucose through a calcineurin/NFAT-independent pathway. On the other hand, no matter what glucose concentration was used in the culture medium, neonatal islets cultured in vitro secreted substantially less insulin than adult islets under 20 mM glucose stimulation. However, supplementing CGA in the culture medium of 11 mM glucose increased the high-glucose-stimulated insulin secretion of the neonatal β-cells (7.29 ± 0.86 ng/ml; Glucose Stimulation Index (GSI) = 22.74 ± 2.68) to a level similar to that of adult β-cells (7.44 ± 0.53 ng/ml; GSI = 27.29 ± 3.29) (*Figure 5B–C*). Taken together, our results demonstrate that optimal glucose could trigger neonatal β-cells functional maturation by reducing low-glucose-induced insulin secretion and increasing high-glucose-stimulated insulin secretion, and the latter mechanism is through the activation of calcineurin/NFAT (*Figure 5A–C*).

Finally, we used mouse islets cultured in vitro to evaluate the correlation between glucose-stimulated calcium transients and GSIS. Compared with adult islets, neonatal islets cultured in medium with 5.6 mM glucose also exhibited a significantly higher ensemble of calcium transients under 2.8 mM glucose (130.3% ± 2.3% versus 102.2% ± 7.1%); these calcium transients were abolished if the neonatal islets were cultured in 11 mM glucose (95.7% ± 10.3% versus 101.2% ± 4.3%), and the neonatal islets became indistinguishable from those of adult islets (*Figure 5D–E* and *Figure 5—figure supplement 1*). Moreover, although including CGA in the culture medium did not reduce low-glucose-triggered calcium transients in neonatal islets cultured in 5.6 mM glucose (126.4% ± 1.9% versus 130.3% ± 2.3%), it did enhance the maximal amplitude of calcium transients triggered by 16.7 mM glucose in neonatal islets cultured both in 5.6 mM glucose (176.4% ± 8.4% versus 147.5% ± 10.2%) and 11 mM glucose (232.2% ± 12.5% versus 181.7% ± 9.5%) (*Figure 5D–F* and *Figure 5—figure supplement 1*). Therefore, glucose-induced calcium response is a reliable and sensitive marker of β-cell functional status.

## Discussion

Visualizing the function of every β-cell in vivo has always been the holy grail for β-cell researchers (*Gotthardt et al., 2014*). Unlike in vivo imaging of β-cells in mammalian species with conventional positron emission tomography and single-photon emission computed tomography that only provide poor spatial-temporal resolution, transparent zebrafish constitutes a unique model that allows high-resolution fluorescent imaging in vivo. In addition to this competitive advantage, we applied the high-resolution 2P3A-DSLM, which has been shown to outperform other microscopes, including the point-scanning TPM, in differentiating individual β-cells in live fish embryos (*Video 1*). With these tools, we achieved the first imaging of individual β-cell function in vivo. We further revealed a heterogeneous functional development of β-cells in vivo and identified its underlying mechanism, that is, islet vascularization regulates glucose delivery to control the heterogeneous functional development of β-cells. Interestingly, there was an increase in optimal glucose concentrations for inducing and enhancing β-cell function from the early to the late hatching period, while a suboptimal or supra-physiological concentration of glucose impaired β-cell functional development (*Figure 3*). Therefore, the precise delivery of the optimal inductive glucose concentrations via blood circulation

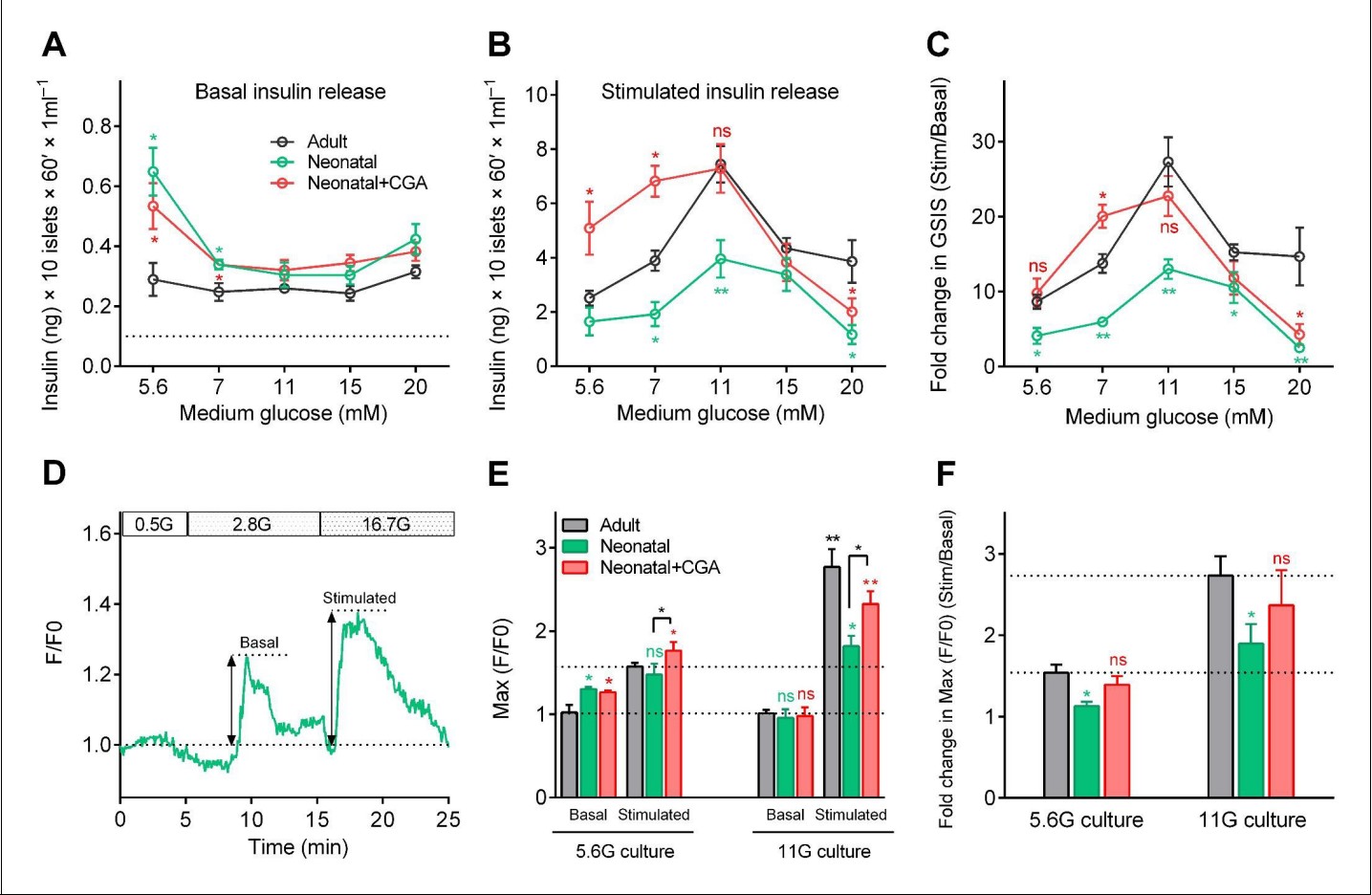

**Figure 5.** Direct activation of calcineurin promoted the optimal maturity of neonatal mouse β-cells in isolated islets in vitro. (A–B) Glucose-induced insulin secretion from neonatal and adult mouse islets sequentially stimulated with 3 mM glucose (A) and 20 mM glucose (B) after isolation from P0 and 8-week-old mice, and for 3 days of culture in different types of media (different concentrations of glucose combined with or without CGA) in vitro. The black dotted line in A indicates the value of the ELISA kit detection limit. n = 4–10 experiments per condition. *p<0.05, **p<0.01; ns, not significant. (C) Glucose Stimulation Index (fold change in GSIS) of the mouse islets described in (A) and (B). n = 4–10 experiments per condition. *p<0.05, **p<0.01; ns, not significant. (D) An illustration of the maximum amplitude of calcium influx in islets under 2.8 mM and 16.7 mM glucose stimulation. (E) Quantification of maximal amplitude of glucose-induced calcium influx in islets under different culture conditions. n = 4–7 islets per condition. *p<0.05, **p<0.01; ns, not significant. (F) Fold change in maximal amplitude of glucose-induced calcium influx in islets under stimulation and resting conditions described in (E). n = 4–7 islets per condition. *p<0.05; ns, not significant. See also *Figure 5—figure supplement 1*.

DOI: https://doi.org/10.7554/eLife.41540.022

The following figure supplement is available for figure 5:

**Figure supplement 1.** Direct activation of calcineurin promoted the optimal maturity of neonatal mouse β-cells in isolated islets in vitro, as indicated by ex vivo islet calcium imaging.

DOI: https://doi.org/10.7554/eLife.41540.023

is critical for the normal development of β-cell function in vivo. An increase in glucose concentration to induce the functional development of β-cells at different stages is likely to be a conserved mechanism in mammals. For example, the plasma glucose level in fetal rodents is relatively low but starts to increase immediately before birth and progressively reaches a plateau (above 4 mM) after P2 (*Rozzo et al., 2009*). From P6 to P20, the plasma glucose level further gradually increases from 6 to 12 mM (*Aguayo-Mazzucato et al., 2006*). This period is accompanied by islet vascularization, the appearance of glucose-responsive β-cells and further maturation (*Hole et al., 1988*; *Reinert et al., 2014*; *Rozzo et al., 2009*). On the other hand, inappropriately elevated blood glucose levels

impaired the β-cell maturation process, as we observed in zebrafish (*Figure 3*). Maternal diabetes in rats leads to immature β-cells in new-borns (*Eriksson and Swenne, 1982*). In human, offspring of type two diabetic women have a higher prevalence of type two diabetes than the offspring of nondiabetic women (*Pettitt et al., 1988*). Therefore, glucose, in addition to being the substrate for adult β-cells, also acts as the critical signaling molecule to induce immature β-cells to become more capable of metabolizing glucose and secreting insulin. This mechanism may represent a general principle that applies to the functional development of other cells.

A previous study from mice indicates that β-cell functional maturation is marked by an increase in the glucose threshold for insulin secretion. However, the mechanisms underlying the increase in the glucose threshold for insulin secretion are not known. Here, through in vivo study in zebrafish and ex vivo study using neonatal mouse islets, we demonstrated that calcineurin/NFAT mediates the inductive roles of glucose, enhancing high-glucose-stimulated insulin secretion during maturation; however, the reduction in low-glucose-stimulated insulin secretion is mediated by a glucose-activated calcineurin/NFAT-independent pathway during neonatal mouse islet maturation. Multiple glucose-activated signaling pathways are potential 'switches' that enhance the glucose metabolism machinery during the development of β-cell function (*Eriksson and Swenne, 1982*; *Lawrence et al., 2002*; *Vanderford et al., 2007*; *Vaulont et al., 2000*; *Zhang et al., 2015*). These pathways may also involve enhancing high-glucose-stimulated insulin secretion. However, which pathway mediates the optimal glucose-induced reduction in low-glucose-stimulated insulin secretion awaits further investigation. Nevertheless, the combined application of a calcineurin activator and optimal glucose overcame the microcirculation deficit in isolated mouse islets and accelerated the ex vivo functional development of neonatal mouse β-cells (*Figure 5*). Similar to isolated mouse islets, stem cell-derived islet-like clusters also lack blood circulation. Different research groups have used a variety of glucose concentrations, ranging from 2.8 to 20 mM, in the final stage of differentiation (*Pagliuca et al., 2014*; *Rezania et al., 2014*; *Russ et al., 2015*); these glucose concentrations may be suboptimal. However, the lack of glucose delivery to all β-like cells within the clusters due to the absence of blood circulation may still be the critical missing factor for promoting high-glucose-stimulated insulin secretion of these cells, even at an optimal glucose concentration, such as 11 mM. Based on our results, the use of a calcineurin activator may overcome this problem and improve the functionality of stem cell-derived β-like cells in clusters. Although the calcineurin/NFAT signaling pathway has been implicated in β-cell development and maturation in mice (*Goodyer et al., 2012*; *Heit et al., 2006*), this pathway has never been manipulated directly in the trials of in vitro differentiation of stem cells into functionally mature β-cells. Our study offers a new strategy of using a calcineurin activator combined with an optimal glucose concentration to promote the functional maturity of stem cell-derived β-like cells within these islet-like clusters in vitro.

Our study also provides a new direction for using in vivo imaging of β-cell mass and function to study mechanisms of islet biology such as regeneration, cell identity or functional changes during disease progression. An understanding of these principals may shed light on methods for developing cell replacement therapies to treat diabetes.

## Materials and methods

### Transgenic zebrafish generation

The Tg (*ins:Rcamp1.07*) reporter zebrafish line was generated using meganuclease-mediated transgenesis as previously described (*Soroldoni et al., 2009*). Briefly, zebrafish BAC_CH211_69I14 (BAC-PAC Resources Center), which contains the zebrafish *insulin* gene, was modified using Red/ET recombineering technology to replace the coding sequence of insulin with *Rcamp1.07* (*Fu et al., 2010*). Rcamp1.07-tagged BAC underwent a second round of recombineering for the subcloning of the modified chromosomal locus into a plasmid backbone containing two I-SceI meganuclease sites. The following primers were used for BAC modification:

Forward primer: 5' ATGTTTTTGATTGACAGAGATTGTATGTGTGTGTTTGTGTCAGTGTGACC CGCCACCATGGGTTCTCATC.
Reverse primer: 5' CCTGTGTGCAAACAGGTGTTTCTGGCATCGGCGGTGGTCAAATCTCTTCA GGCAGATCGTCAGTCAG.

For subcloning:

Forward primer: 5' AGTCCATTAAATAATATCTTGTAGAATTATGTTTTTAAAAAGTACCAATGCC GTAGGGATAACAGGGTAATTTAAGC.
Reverse primer: 5' CAACTTTTTCACAAACACTGACCAAAACAAGCTACATGTTTTAGAGGCATTA GGGATAACAGGGTAATTGCACTG.

The resulting constructs were co-injected with I-SceI meganuclease (Roche, Indianapolis, IN) at a DNA concentration of 100 ng/μl into one-cell stage zebrafish embryos. These F0 embryos were screened for the transient expression of Rcamp1.07 in the pancreatic islet at 48 hpf using a fluorescence stereomicroscope. The positive F0 founders were raised to adulthood and screened by visual inspection of their F1 progenies from outcrossing with the wild-type AB strain. Based on the intensity of the fluorescence signal, one founder was selected, and subsequent generations were propagated and expanded.

The Tg (*ins:EGFP-GSG-T2A-dn-zCnA*) zebrafish line was generated using Tol2 transposase RNA-mediated transgenesis as previously described (*Kawakami, 2007*). Briefly, *dn-zCnA* was constructed by deleting the autoinhibitory and the calmodulin-binding domains through introducing a stop codon at the N396 amino acid and by mutating the histidine at the position 152, a phosphatase-active site, to glutamine (*Zou et al., 2001*). The GSG-T2A peptide was used to separate EGFP and dn-zCnA elements. The EGFP-GSG-T2A-dn-zCnA fragment was cloned downstream of 2.1 kb of the proximal insulin promoter and into a Tol2 plasmid using a ClonExpress system. The final construct was injected along with Tol2 transposase RNA into Tg (*ins:Rcamp1.07*) eggs to generate mosaic Tg (*ins:EGFP-GSG-T2A-dn-zCnA*);Tg (*ins:Rcamp1.07*) F0 fish for imaging analysis.

## Zebrafish care and handling

The wild-type AB strain and transgenic fish were maintained and handled according to the institutional guidelines of animal usage and maintenance of Peking University. The Tg (*ins:EGFP*) fish were from Dr. Lin Shuo at UCLA; the Tg (*flk1:mCherry*) fish were from Dr. Zhang Bo at PKU; the Tg (*flk1:GFP*) fish were from Dr. Chen Jau-Nian at UCLA and the Tg (*elavl3:Gcamp6s*) fish were from Dr. Florian Engert at Harvard University. Tg (*ins:Rcamp1.07*) was crossed with heterozygous *cloche$^{-/+}$* to obtain Tg (*ins:Rcamp1.07*);*cloche$^{-/+}$* zebrafish. Tg (*ins:Rcamp1.07*);*cloche$^{-/-}$* embryos from the crossing of Tg (*ins:Rcamp1.07*);*cloche$^{-/+}$* with *cloche$^{-/+}$* fish were used for the imaging experiments. Phenylthiourea (0.002%) (PTU, Sigma, St. Louis, MO) was added at 12 hpf to prevent pigment synthesis. Prior to live imaging experiments, embryos were anesthetized with 0.01% tricaine (Sigma).

## Pharmacological treatment

Heterozygous Tg (*ins:Rcamp1.07*) embryos were used for pharmacological treatments and imaging analyses. All chemicals were prepared as high-concentration stocks and diluted in E3 medium to the final concentrations used for treatment, which were carefully selected to be nontoxic and effective. Embryos were treated with each chemical for 9 hr, either from 36 to 45 hpf, 44–53 hpf or from 60 to 69 hpf, followed by a 3 hr recovery after the chemicals were completely washed out with E3 medium to test the pharmacological effects of the chemicals on the functional development of pancreatic β-cells. Imaging analyses for functional evaluation were performed at 48 hpf, 56 hpf or 72 hpf. For the pharmacological treatment of zebrafish embryos, 10 mM 2,3-BDM (Sigma) was used to block blood circulation. 2,3-BDM is a well-characterized, low-affinity, noncompetitive inhibitor of skeletal muscle myosin-II that blocks myofibrillar ATPase in a dose-dependent manner. This compound decreases the myocardial force, thereby abolishing blood flow at the 10 mM concentration in our experiments (*Bartman et al., 2004*). Zebrafish embryos were treated with 3 mM 3 MPA (Santa Cruz Biotechnology, Dallas, Texas) to inhibit endogenous glucose production, 141.2 μM chlorogenic acid (CGA, Sigma) or 10 μM FK506 (Invivogen, San Diego, CA) to activate or inhibit calcineurin, respectively, and 2.5 μM ProINDY (Tocris, Minneapolis, MN) or 20 μM VIVIT peptide (Tocris) to activate or inhibit NFAT, respectively.

## Uptake of the fluorescent D-glucose analog by live zebrafish embryos

2-NBDG (Invitrogen, Waltham, MA) is a fluorescent deoxyglucose analog that can be taken up by the cells through glucose transporters. Fluorescence generated by 2-NBDG is therefore used to measure glucose uptake with a fluorescence microscope, as its intensity is proportional to glucose

uptake by the cells (*Yamada et al., 2007*). 2-NBDG stock was diluted to working concentrations and then ready to be used for incubating live zebrafish embryos. After thoroughly washing, images were captured under a TPM (Zeiss 710). The 2-NBDG signal was excited at 920 nm and collected between 510 and 540 nm.

## Spinning-disc confocal and 2P3A-DSLM imaging

For the confocal time-lapse imaging, images were captured using an Olympus spinning-disc confocal microscope with a 10×/0.4 objective and 1.6 × preamplifier. Tg (*ins:Rcamp1.07*) embryos were embedded in a 1% ultra-pure agarose (Invitrogen) cylinder and immersed in an imaging chamber filled with E3 medium containing 0.01% tricaine. A D-glucose stock solution was added to the E3 medium to a final concentration of 20 mM during stimulation. Time-lapse images were captured once per second with a 100 ms exposure time. Images were collected using MetaMorph software and analyzed with Fiji software.

High-resolution images of pancreatic β-cells and blood vessels in live zebrafish embryos were captured with a 2P3A-DSLM equipped with two 40×/0.8 water lenses, as previously described (*Zong et al., 2015*). Briefly, anesthetized embryos were embedded in a 1% ultra-pure agarose cylinder and immersed in E3 medium containing 0.01% tricaine. When monitoring β-cell function within zebrafish embryos, glucose was added to the E3 medium to a final concentration of 20 mM as an acute stimulation. For the 2D time-lapse imaging experiments used in the statistical analysis, the islet was optically sectioned into 5–6 layers to ensure that the calcium transients were recorded in all β-cells within the islet. For the fast volumetric imaging and reconstruction of calcium transients within the whole islet, the islet was optically sectioned into 25 layers. Each layer was captured five times with an 8 ms exposure time and was averaged as one single image. Images were collected using the HCImage software (Hamamatsu) and processed with R-L deconvolution by the Fiji software. The volumetric calcium transients were reconstructed using Amira software (FEI). For every β-cell, the relative change in the fluorescence intensity of Rcamp1.07 was indicated by the $\Delta F/F0$. The $\Delta F$ value was calculated as the difference between the absolute intensity values (F) at any time point and the initial intensity value (F0) at the first time point.

## Comparison among 1P-SPIM, TPM and 2P3A-DSLM in 3D imaging of islet in vivo

For 1P-SPIM imaging, we used a homemade SPIM setup equipped with a 40×/0.8 water lens. For TPM imaging, we used a fast resonant-scanned TPM equipped with a 40×/0.8 water lens. The same 72 hpf zebrafish samples were sequentially imaged with 1P-SPIM, TPM and 2P3A-DSLM. Under each configuration, the whole islet was optically sectioned by 100 planes (z-step: 500 nm) with an exposure time of 150 ms per frame. Images were captured using the HCImage software and processed with R-L deconvolution using Fiji software.

## Cryostat sectioning and immunofluorescence

The embryos were fixed in 4% paraformaldehyde (PFA, AppliChem, St. Louis, MO) at 4°C overnight. After washing, the embryos were dehydrated in 30% sucrose (Sigma) and then transferred to embedding chambers filled with OCT compound (Tissue Tek, Sakura Finetek, Torrance, CA). After embedding, the samples were frozen in liquid nitrogen as soon as possible. Sectioning was performed using a Leica CM1900 Cryostat set to a 10 μm thickness and a −25°C chamber temperature. The sections were collected and kept at −20°C in a sealed slide box. Each section was gently transferred to room temperature (RT) by allowing the section to melt onto the slide for at least 1 hr prior to immunostaining. Sections were washed once and then permeabilized with acetone for 5–10 min at RT. After extensively washing and blocking with PBST (PBS + 0.1% Tween-20) containing 0.2% bovine serum albumin (BSA, AppliChem) and 5% fetal bovine serum (Gibco, Gaithersburg, MD) for 1 hr at RT, the sections were incubated with primary antibodies overnight at 4°C. Then, secondary antibodies were applied at 4°C overnight after thorough washing. For nuclear staining, the sections were incubated with 2 μg/ml DAPI (Solarbio, Beijing, CN) for 10 min. After extensive washing, the sections were mounted in 80% glycerol (Sigma) and imaged under an Olympus spinning-disc confocal microscope. The primary antibodies included monoclonal rat anti-mCherry antibody (1:200, Thermo, Waltham, MA) used to detect Rcamp1.07 and polyclonal guinea pig anti-insulin antibody

(1:200, Dako, Carpentaria, CA). The secondary antibodies included Alexa Fluor 568 goat anti-rat IgG (1:500, Thermo) and DyLight 488 goat anti-guinea pig IgG (1:500, Thermo).

## In vitro culture of mouse islets, measurement of GSIS and calcium imaging

Adult islets from 8-week-old mice were isolated as previously described (*Wang et al., 2016*). For islet isolation from P0 mice, pancreata were directly dissected without perfusion and digested with 0.5 mg/ml Collagenase P (Roche). The isolated islets were cultured for 3 days in RPMI1640 media containing different concentrations of glucose (5.6 mM, 7 mM, 11 mM, 15 mM and 20 mM) with or without CGA (56.48 µM). The culture medium was changed daily.

To measure GSIS, 10 islets of similar sizes were selected and preincubated in KRB buffer for 2 hr at 37°C in a 5% $CO_2$ incubator. The islets were then transferred to low-glucose (3 mM) KRB buffer and incubated for 1 hr at 37°C, 5% $CO_2$. The supernatant was collected, and basal insulin secretion was measured. The same islets were transferred to high-glucose (20 mM) KRB buffer for another 1 hr incubation at 37°C, 5% $CO_2$. The supernatant was collected and stored at −20°C, and later, the insulin content was measured using the rat/mouse insulin ELISA kit (Millipore, St. Louis, MO).

For ex vivo calcium imaging, the islets were washed with KRB buffer, incubated with 10 µM Cal-520 AM (AATB) for 60 min, washed twice with KRB buffer, and then incubated further at 37°C for another 15 min without the dye. Before imaging, islets were attached to chamber coated with Cell-Tak (Corning, NY, USA) and then immediately staged on spinning-disc confocal microscope for the acquisition of high-resolution images. Time-lapse images were captured once per second at a single-cell resolution of 40 × magnification. The progression of glucose challenges and the time of stimulation during imaging were as follows: 5 min in 0.5 mM glucose KRB buffer; 10 min in 2.8 mM glucose KRB buffer; and 10 min in 16.7 mM glucose KRB buffer. Images were analyzed using Fiji software.

## Method used to map the data presented in *Figure 1D* and *Figure 2A*

Based on both 2D images and 3D reconstructed images of Tg (*ins:Rcamp1.07*) zebrafish islets, all β-cells were manually outlined according to their localization (inside or outside, *Figure 2A*) or glucose responsiveness (responsive or nonresponsive, *Figure 1D*). The coordinate positions of the β-cells were added to the region of interest (ROI) manager in the Fiji software. Then, we loaded the raw data and the ROI information into our self-developed MATLAB program and transformed the grayscale map to hue-saturation value (HSV) color space, setting saturation to one and the proportionate value to the gray level distribution. The hue consists of 0, 120 and 240 representing three colors. As shown in *Figure 2A* and *Figure 1D*, the inside or glucose-responsive cells are red colored (hue = 0), and the outside or glucose nonresponsive cells are blue colored (hue = 240) or green colored (hue = 120).

## Statistical analyses

All data were analyzed using GraphPad Prism six software. The results are displayed as the mean value ±standard error of the mean (SEM). Unpaired Student's two-tailed t-tests were used to compare data between two indicated groups. One-way ANOVA followed by Dunnett's test was used for multiple comparisons with the control group. The asterisks *, ** and *** indicate significance with *p* values less than 0.05, 0.01 and 0.001, respectively.

## Study approval

Animal care, generation of transgenic zebrafish lines, in vivo imaging of living zebrafish embryos and all other experiments involving zebrafish and mouse islets were approved by the IACUC of Peking University in China (reference number: IMM-ChenLY-2).

## Acknowledgments

We thank Dr. Lin Shuo, Dr. Zhang Bo, Dr. Chen Jau-Nian, Dr. Liu Feng and Dr. Florian Engert for sharing the fish lines with us. We thank Dr. Liao Bo-Kai, Ms. Zheng Ruyi and Ms. Meng Liying for technique assistant. The work was supported by the grants from the National Science and

Technology Major Project Program (2016YFA0500400), National Natural Science Foundation of China (91854112 and 91750203, 31327901, 31521062, 31570839, 31301186), and Beijing Natural Science Foundation (L172003).

## Additional information

### Funding

| Funder | Grant reference number | Author |
|---|---|---|
| National Science and Technology Major Project Program | 2016YFA0500400 | Liangyi Chen |
| National Natural Science Foundation of China | 91854112 | Yanmei Liu |
| National Natural Science Foundation of China | 91750203 | Yanmei Liu |
| National Natural Science Foundation of China | 31327901 | Liangyi Chen |
| National Natural Science Foundation of China | 31521062 | Liangyi Chen |
| National Natural Science Foundation of China | 31570839 | Liangyi Chen |
| National Natural Science Foundation of China | 31301186 | Yanmei Liu |
| Beijing Natural Science Foundation | L172003 | Liangyi Chen |

The funders had no role in study design, data collection and interpretation, or the decision to submit the work for publication.

### Author contributions

Jia Zhao, Conceptualization, Resources, Data curation, Software, Formal analysis, Validation, Visualization, Writing—original draft, Writing—review and editing; Weijian Zong, Jiayu Shen, Resources, Software, Formal analysis, Validation, Methodology; Yiwen Zhao, Resources, Data curation, Validation, Visualization; Dongzhou Gou, Resources, Data curation, Software, Validation, Visualization; Shenghui Liang, Conceptualization, Resources, Data curation, Software, Formal analysis, Validation, Investigation, Visualization, Methodology; Yi Wu, Resources, Data curation, Software, Formal analysis, Validation, Methodology; Xuan Zheng, Runlong Wu, Resources, Data curation, Software, Formal analysis, Validation, Visualization; Xu Wang, Fuzeng Niu, Resources, Software, Methodology; Aimin Wang, Yunfeng Zhang, Conceptualization, Supervision, Investigation, Methodology; Jing-Wei Xiong, Conceptualization, Resources, Supervision, Investigation, Methodology; Liangyi Chen, Conceptualization, Resources, Data curation, Software, Formal analysis, Supervision, Funding acquisition, Validation, Investigation, Methodology, Writing—original draft, Project administration, Writing—review and editing; Yanmei Liu, Conceptualization, Resources, Data curation, Software, Formal analysis, Supervision, Funding acquisition, Validation, Investigation, Visualization, Methodology, Writing—original draft, Project administration, Writing—review and editing

### Author ORCIDs

Jia Zhao  http://orcid.org/0000-0002-1669-6992
Liangyi Chen  http://orcid.org/0000-0003-1270-7321
Yanmei Liu  http://orcid.org/0000-0001-9380-2560

### Ethics

Animal experimentation: Animal care, generation of transgenic zebrafish lines, in vivo imaging of the live zebrafish embryos and all other experiments involving zebrafish and mouse islets were approved by the IACUC of Peking University in China (reference no. IMM-ChenLY-2).

### Decision letter and Author response
Decision letter https://doi.org/10.7554/eLife.41540.035
Author response https://doi.org/10.7554/eLife.41540.036

## Additional files

### Supplementary files
• Supplementary file 1. Comparison of the critical events and the time windows in zebrafish and mouse pancreas development. The table shows comparison of developmental process and critical stage in zebrafish and mouse pancreas development.
DOI: https://doi.org/10.7554/eLife.41540.024
• Transparent reporting form
DOI: https://doi.org/10.7554/eLife.41540.025

### Data availability
All data generated or analysed during this study are included in the manuscript and supporting files.

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
