## [Decision Letter]

[Editors’ note: a previous version of this study was rejected after peer review, but the authors submitted for reconsideration. The first decision letter after peer review is shown below.]

Thank you for submitting your work entitled "In Vivo Imaging β-cell Function Reveals Two Waves of β-cell Maturation" for consideration by *eLife*. Your article has been reviewed by three peer reviewers, and the evaluation has been overseen by a Senior/Reviewing Editor. The reviewers have opted to remain anonymous.

Our decision has been reached after consultation between the reviewers. Based on these discussions and the individual reviews below, we regret to inform you that your work will not be considered further for publication in *eLife*.

This manuscript describes experiments where β-cells within the developing zebrafish embryo are imaged using a two-photon light-sheet microscope. The stated rationale for doing so is the ability of monitoring β-cell activity in situ in a non-invasive manner. While these are technically demanding experiments, the spatial and temporal resolution of the image data is limited compared to what has been recently reported using more conventional microscopes in an ex vivo setting. The novel insight that is afforded by this technique and is therefore limited and while the choice for zebrafish as a model is transparent and well-justified, it inevitably raises questions regarding the degree of similarities between zebrafish and rodent β-cells including if β-cells in the zebrafish islet are coupled via gap junctions (the videos provided suggest not?), what the glucose threshold is for a zebrafish β-cell (what is non-fasting plasma glucose in a zebrafish anyway). Also, the zebrafish islet in the paper is divided into a mantle and core, but with only 15-20 β-cells making up the islet, it is questionable if there truly is a mantle/core distinction such as the one that is readily apparent in rodent islets. The visualization of the vascularization of the islet during development is nice but would not have required a two-photon light-sheet setup.

The major concern is that throughout the paper, β-cell maturation and function are conflated. This betrays a serious flaw in the premise of the work that pervades the entire paper. Transgenic zebrafish are used that express the Rcamp red calcium indicator in β-cells. These fish are exposed to glucose and an Rcamp response to glucose is interpreted to indicate functional maturity of the responding β-cells. Using this approach, the investigators observe calcium responses first on the outside of the islet followed by β-cells in the islet core and conclude from this that β-cells mature from mantle to core. The problem with this interpretation is that if β-cells at the core were mature but were not exposed to glucose, they would not respond with a calcium response. Indeed, the investigators then demonstrate the vascularization proceeds from the mantle to the core and suggests that the mantle β-cells respond first simply because they are exposed to the circulation earlier in development. The experiments where vascularization is perturbed chemically or via genetic perturbation predictably cause reduced β-cell functionality, but likely not because β-cell maturation is inhibited, but because glucose exposure of functionally mature β-cells is limited due to the lack of circulation. There is an attempt of demonstrating that mature β-cells exist in islets using a MafA antiserum, but there is no evidence that this antiserum works, let alone detects zebrafish MafA despite the considerably evolutionary distance between fishes and mammals (beyond the manufacturers claims it may, which is clearly insufficient validation). It is telling that the images of whole mount MafA staining appear to be cytoplasmic and do not overlap well with insulin, while MafA is a nuclear transcription factor of β-cells that typically requires antigen retrieval for detection.

A similar problem arises for the experiments where calcineurin/NFAT is inhibited. Doing so inhibits calcium responses in vivo and is interpreted as a reduction in β-cell maturity. The possibility that blockade of calcineurin/NFAT blocks β-cell function is not considered, yet likely.

After suggesting that maturation of β-cells starts at the mantle and progresses to the core, this model is transplanted to mouse β-cells. The rationale is that 'β-cells of neonatal mouse islets respond poorly to glucose and need more than one week to mature in vivo (Blum et al., 2012).' Actually, this paper by Blum showed that β-cells in the first days post-partum are *more* sensitive to glucose and that their glucose threshold increases as the islets mature. This was subsequently shown by others to be caused by the onset of intra islet feedback on β-cell function. It appears that the experiments on mouse islets in this current manuscript are based on only a handful of the pertinent papers that have either not been carefully read or have been mis-understood.

To demonstrate that maturation proceeds from mantle to core in mouse islets, the investigators assay glucose uptake using a fluorescent glucose tracer. They observe highest uptake in the outer layers of the islet and appear to interpret this as evidence in support for a model where maturation proceeds from the mantle to the core? But these islets are done in 8-week old islets, when the entire β-cell mass has long since matured. There is no evidence that the cells that take up glucose tracer at the periphery are actually β-cells, as the mouse islet mantle is the site of α- and δ-cells. I could not find any indication to rule out that the selective labeling of the mantle is caused by the limited diffusion of the tracer into the islet core.

The effects of calcineurin on zebrafish β-cell function and maturation appear to confirm published work from over a decade ago. The novelty of these findings may be limited to the zebrafish pancreas field.

Additional comments from the reviewers:

1) In several places in the text, the authors refer to β-cell proliferation. However, there are no assays actually performed to assess β-cell proliferation. Either the authors need to perform these experiments, or they need to remove such wording from the manuscript. They can refer to an increase in β-cell number, but in the absence of any direct evidence, one could not state with certainty whether this was from proliferation or neogenesis.

2) The videos should be annotated to help the viewer. There were many times that I did not know what I was looking at, did not know when glucose had been added, if at all, and could not tell what the changes were that I should be focusing on.

3) In Figure 3C, the authors show that 20 mM glucose is *not* the optimal glucose concentration for the effects they are interested in. In fact, 8 mM glucose seems to be better. Why then do they use 20 mM glucose for most experiments?

4) In the Discussion, the authors refer to "locally synthesized glucose" as initiating β-cell maturation. What does this mean? What is the source of glucose? Along those lines, since feeding has not yet begun at 72 hpf, and the source of nutrition is the yolk, how do the authors correlate β-cell maturation in the fish with the changes noted in mice at the time of nursing initiation and switching to chow at weaning?

5) The authors have not "discovered" heterogeneity in β-cell maturation (Discussion, second paragraph). Their data do nicely support previous studies showing β-cell maturation heterogeneity looking at Ucn3 and MafA expression. Please cite these previous studies appropriately.

6) The authors refer incorrectly to previous studies of islet vascularization in the mouse (Discussion, second paragraph). They are referred to Brissova et al. (2006, Diabetes) which shows very nicely that blood flow precedes islet morphogenesis. How do they interpret their findings in light of this previous study?

7) There are several data sets for which a t-test is not appropriate. Any time multiple treatments are being compared, an ANOVA should be used. Please consult with a statistician.

8) In Figure 1—figure supplement 2, what is the rationale for the time points examined? Waiting 3 minutes until after glucose administration to start collecting the data seems too long. Likewise, taking an image once a minute seems too far apart. I am concerned that the authors missed something.

9) Are the VE+ cells the same cells that are Glu+ (Figure 2B). How do the authors define VE β-cells?

10) In Figure 2C, what are the holes? α-cells, β-cells not expressing the reporter? Other endocrine cells? Why are there some cells outlines with no color assigned?

11) It needs to be made clearer that Figure 5A-C is from islets from 8-week-old mice, not neonates.

[Editors’ note: what now follows is the decision letter after the authors submitted for further consideration.]

Thank you for submitting your article "In Vivo Imaging of β-cell Function Reveals Glucose-mediated Heterogeneity of β-cell Functional Development" for consideration by *eLife*. Your article has been reviewed by three peer reviewers, and the evaluation has been overseen by Marianne Bronner as the Senior and Reviewing Editor.

The reviewers have discussed the reviews with one another and the Reviewing Editor has drafted this decision to help you prepare a revised submission. The most essential revisions are summarized below, but we refer you to the individual reviewer comments for further details.

Summary:

Here, the authors use light-sheet microscopy to image the pancreatic β-cells in zebrafish and in cultured mouse cells. While conventional confocal and light-sheet microscopy is not adequate for their functional imaging, their enhanced multi-photon light-sheet is capable of recording with the needed resolution. They find a number of changes over time and heterogeneities in the level and synchrony of intracellular calcium transients and also address the potential role of glucose, calcineurin and NFAT in the development of responsive β-cells.

Essential revisions:

1) It is critical to address possible off target effects of the pharmacological reagents. A control experiment needs to be added showing that the calcium levels of a totally unrelated cell type (e.g. a CNS neuron) are not affected as observed in pancreas cells.

2) The dominant negative experiments are not convincing. If the perturbing dominant negative construct does nothing more than make the cells thrive a little less, the obtained result (slowing the appearance of marker in the expressing cells) would be predicted to result. This is likely an artifact that needs to be carefully controlled.

3) The mouse studies are incomplete and seem to have been performed with inadequate attention to standardization. The Glucose Simulated Insulin Release (GSIS) studies are not presented in the text of the Results and in Figure 5. The calcium imaging and GSIS are performed at different concentrations.

4) The finding should be more completely presented and fully addressed in the text and with appropriate statistical treatments.

*Reviewer #1:*

In this manuscript, the authors use light-sheet microscopy to image the pancreatic β-cells in zebrafish and in cultured mouse cells. They show that conventional confocal and light-sheet microscopy is not adequate for their functional imaging, but that their enhanced multi-photon light-sheet is capable of recording with the needed resolution.

The authors find a number of changes over time and heterogeneities in the level and synchrony of intracellular calcium transients. They argue that the zebrafish β-cells mature from the mantle to the core, but become indistinguishable by 72 hours. Imaging the labeled vasculature showed a similar progression, so the mantle-core difference is likely due to vascularization. Blocking the heartbeat suggests this is due to the circulation rather than the vascular endothelium.

The authors address the potential role of glucose in the development of responsive β-cells, and offer several findings that are interesting, but seem incomplete. The studies showing the delivery of labeled glucose analogs show an apparent non-linearity that suggests there are unknowns not being addressed. A dose of a labeled glucose analog that is 2.5 times higher completely penetrates the core; the smaller dose completely fails, as described. The concentrations used in the glucose studies seem to range widely, from physiological to toxic.

The molecular studies on calcineurin and NFAT offer some interesting findings, that suggest involvement, but the results are presented in a fashion that makes it hard to view the issue as settled:

"The first glucose responsive β-cells that appeared at 48 hpf were all EGFP negative, indicating that *dn-zCnA* prevents β-cells from acquiring glucose responsiveness."

If the perturbation prevented the glucose responsiveness, it would not just be the first cells that are perturbed. Slowing could result from a variety of issues that are non-specific, just from expressing the perturbing transgene.

The mouse studies are incomplete and seem to have been performed with inadequate attention to standardization. The Glucose Simulated Insulin Release (GSIS) studies are not presented in the text of the Results and in Figure 5, the calcium imaging and GSIS are performed at different concentrations.

This study has many interesting aspects to it, and it is clear that the approaches being deployed will offer important insights. It is within the authors' grasp to create a convincing and complete analysis of each of the findings they present, without any heroics or development of new technologies.

The present version of the manuscript is a patchwork of incomplete studies that do not resolve any of the exciting aspects the study offers access to. I found the manuscript unconvincing, and incompletely presented. For example, it offers descriptions of results that are incompletely presented or misleading (see the comments on the penetration of labeled glucose analogs, and on the *dn-zCnA* studies).

I would strongly recommend that the present version be rethought, and focus on presenting an analysis of either the zebrafish or the mouse, creating a substantial contribution to the literature.

*Reviewer #2:*

These workers use the zebrafish islet model wherein β-cells were tagged Rcamp to monitor β-cell functional maturation, differences in the islet core vs. periphery, and how this is further modulated by vascularization and maturation process by calcineurin-NFAT signaling. I do think that the microscopy approaches they developed and using the zebra fish islet model, is a great strategy to track the spatio-temporal islet developmental biology. This is important to the overall islet biology community. The findings of islet β-cell functional maturation, requirement for vascularization and calcineurin were sufficiently novel, and used to test this model strategy,

*Reviewer #3:*

The paper by Zhao et al. provides an in vivo analysis of developing pancreatic cells which is particularly challenging given that these cells are deep within the embryo.

1) Their comparisons of different microscopes convincingly showed that their 2P3A-DSLM provided the best images. I see that they deconvolved their two-photon light-sheet data. Did they deconvolve the other data as well? One thing I could not determine from their methods was if they rotated the embryo to collect multiple views of each embryos to improve their Z-resolution (as in the classic SPIM paper)? This would also make their data more isometric. I think their spinning disc confocal data would have looked better with a higher magnification lens that the 10x they used.

2) Their finding that glucose responsive β-cells appeared in vivo earlier than previously reported (48 hpf instead of 52 hours) was very interesting.

3) The increase in calcium activity from the mantle to the core was another interesting finding. One question I had about the increase in calcium signaling from 56 to 72 hpf was whether these were the same cells or new cells with increased activity. Their data should easily reveal this, and it is an important point for them to mention.

4) Their hypothesis that blood vessels initiated β-cell functional development as measured by calcium activity may be true but the data from the cloche mutant and their pharmacological reagents (e.g. BDM) doesn't exclude another hypothesis, that β-cell functional maturation may cause these cells to secrete a factor that promotes angiogenesis. Remember blocking circulation early did not affect β-cells' ability to acquire glucose responsiveness. I am glad that the authors went beyond the correlation to actually test their hypothesis though it seems like their conclusions are still based more on correlation than causation.

5) I felt that the authors data from the cloche mutant and their dnCalcineurin line were more convincing than their data from pharmacological reagents for which I'm always concerned about off target effects. How specific are these particular reagents? The authors say they did careful dose response curves but the control I would really like to see is if the drugs affected calcium signaling in a totally unrelated cell like a CNS neuron. Do calcium levels change in these cells with drug treatment? I would be more convinced by data from Calcineurin or NFAT mutants. I'm not fond of calling calcineurin/NFAT signaling a "master regulator" as they did in Figure 4. The authors state that calcineurin/NFAT signaling is a major pathway for glucose mediated β-cell development but that didn't seem to be the case for outer cells at 56 hpf?

6) I'm a little surprised that the data in Figure 3 did not show larger differences. I do agree that the 20 mM Glucose concentration is likely toxic and so it shouldn't be used.

7) For the mouse data the authors used 20 mM Glucose level to activate insulin. Is this concentration not toxic for mouse islet cells? I was also a little confused by the calcineurin/NFAT independence at low glucose level but dependence at high glucose levels. In the Discussion it was a little difficult to compare the mouse and fish data because insulin levels were only measured in the mouse. For mouse cells they had to use a drug since they didn't have the dnCalcineurin transgenic they used in fish.

[Editors' note: further revisions were requested prior to acceptance, as described below.]

Thank you for resubmitting your work entitled "In Vivo Imaging of β-cell Function Reveals Glucose-mediated Heterogeneity of β-cell Functional Development" for further consideration at eLife. Your revised article has been favorably evaluated by Marianne Bronner (Senior Editor) and 2 of the original reviewers.

The manuscript has been improved but there are few remaining minor issues that need to be addressed before acceptance, as outlined below

Required changes:

1. The response to reviewer #3, point 4 is great but it's not in the paper. Please include in the manuscript a sentence or two from the response to this reviewer. Same for reviewer #3, point 6. Why not mention that a living organism may have some buffering mechanism in the relevant part of the paper? That would be a nice Discussion point.

2. In the figure legends, it might be good to mention that inside = core and outside = mantle. In the text they only use the terms core and mantle which is appropriate.

3. Subsection “Fine glucose concentrations regulate the heterogeneous development of β-cell function in vivo”, first paragraph: “periphery” should be “peripheral”.

---

## [Author Response]

[Editors’ note: the author responses to the first round of peer review follow.]

We thank all of the reviewers for evaluating our work and providing insightful comments. We have carefully designed and done many new experiments per your comments and suggestions. Now we have fully addressed all of your concerns, demonstrated the conserved glucose-induced calcineurin/NFAT-dependent β-cell maturation in both zebrafish and mouse, and revealed that a novel, glucose-induced but calcineurin/NFAT-independent process also contributed to β-cell maturation in mouse. In the completely revised manuscript, now entitled “In vivo imaging of β-cell function reveals glucose-mediated heterogeneity of β-cell functional development”, the following novelties and major advances are detailed:

1) We have demonstrated that β-cells that do not show a calcium response have access to the transient increase of glucose under high glucose incubation.By using fluorescent deoxyglucose analog 2-NBDG to incubate fish embryos, we demonstrated that physiological dose of glucose (8 mM) did not penetrate islet core without microcirculation (Figure 2—figure supplement 3). The lack of the chronic induction of physiological dose of glucose led to defective β-cell functional development in islet core from 72 hpf cloche mutants and BDM-treated embryos (Figure 2F). On the other hand, supraphysiological dose of 2-NBDG (20 mM) readily penetrated islet core even in cloche mutants without any circulation (Figure 2—figure supplement 3). Functionally, defective β-cells in the islet core of cloche mutants could be rescued by direct activation of calcineurin, again indicating that these cells can be accessed by high glucose (Figure 4C).Therefore, islet microcirculation is essential for chronically delivering physiological dose of glucose to β-cells in the islet core for their functional development in vivo, but does not prevent their function to be evaluated under acute stimulation with supraphysiological dose of glucose.

2) Not only have we figured out the reason of the difference between Blum’s data and our previous data, but also unraveled two mechanistic different functional maturation processes of β-cells, including a previously unappreciated one.By systematically examining insulin secretion of neonatal and adult mouse islets cultured under different concentrations of glucose (Figure 5), we found that β-cell maturation ex vivo composes of two processes: the reduction of low glucose-stimulated insulin secretion and the increase of high glucose-stimulated insulin secretion. While calcineurin/NFAT participates in prompting high glucose-stimulated insulin secretion, glucose also activates other downstream pathways to reduce low glucose-stimulated insulin secretion.

3) Per your suggestion, we have systematically evaluated four commercially available MafA antibodies that target conserved epitopes in zebrafish, and found none of them labeled nucleus structures of β-cells in adult fish. Therefore, we have completely removed all MafA-related data from the manuscript. On the other hand,the variable functions of neonatal mouse islets cultured under different concentrations of glucose speaks for the importance of being able to image β-cell function in vivo. By displaying that glucose-induced calcium responses correlated nicely with glucose-stimulated insulin secretion in mouse neonatal islets cultured in vitro(Figure 5), we further demonstrate that glucose-induced calcium response is a reliable and sensitive marker of β-cell function. Moreover, the mechanisms of functional development of β-cells revealed by visualizing glucose-induced calcium responses from individual β-cells in live zebrafish embryos were confirmed in mouse islets, reinforcing the validity of using glucose-induced calcium responses as a marker of β-cell function status.

This manuscript describes experiments where β-cells within the developing zebrafish embryo are imaged using a two-photon light-sheet microscope. The stated rationale for doing so is the ability of monitoring β-cell activity in situ in a non-invasive manner. While these are technically demanding experiments, the spatial and temporal resolution of the image data is limited compared to what has been recently reported using more conventional microscopes in an ex vivo setting.

We thank you for the appreciation of our technology as “non-invasive” and “technically demanding”. Because the islets within live zebrafish we imaged did not have a fine structure, our high-resolution two-photon light-sheet microscope seemed to have a resolution inferior to the imaging under ex vivo setting. However, we were able to resolve calcium transients from single β-cells in live fish in vivo (Videos 2-3 and 5-6), which can only be achieved in isolated islets from zebrafish older than 4 dpf as shown by others (Singh, Janjuha et al., 2017). As we have shown that the function of isolated islets are susceptible to their culture conditions (Figure 5), we argue that to be able to image β-cell function in vivo offers a clear advantage as compared to ex vivo imaging, which becomes possible due to our innovation in the microscope itself.

The novel insight that is afforded by this technique and is therefore limited and while the choice for zebrafish as a model is transparent and well-justified, it inevitably raises questions regarding the degree of similarities between zebrafish and rodent β-cells including if β-cells in the zebrafish islet are coupled via gap junctions (the videos provided suggest not?), what the glucose threshold is for a zebrafish β-cell (what is non-fasting plasma glucose in a zebrafish anyway).

We thank you for the reminder. We argue that β-cells from zebrafish indeed share conserved mechanisms with those from rodents based on four points listed below:

1) The composition of cell types and the structural organization of pancreas in zebrafish are similar to mammalian (Prince, Anderson et al., 2017). This is also why the knowledge from zebrafish developmental biology research helps in establishing protocols for differentiating embryonic stem cells into β-cells.

2) Following your suggestion, we did observe synchronized Ca^2+^ signals among neighboring β-cells in some islets of 72 hpf zebrafish (Figure 1—figure supplement 4D-E). This data implies that β-cells within zebrafish islets are connected by gap junction, which may become even more prominent as the fish matures. We have added this information in our revised manuscript (Figure 1—figure supplement 4D-E, Video 2 and subsection “Visualization of embryonic β-cell function in vivo using 2P3A-DSLM”, last paragraph).

3) The fasting plasma glucose measured in adult zebrafish is about 2.8 mM, while the peak of the OGTT (after oral administration of 1.25 mg glucose/g body weight) reaches around 8.3 mM (Zang, Shimada et al., 2017). These values are also comparable to the normal range of plasma glucose fluctuations in rodents.

4) Finally, the calcineurin/NFAT-dependent β-cell maturation process in zebrafish also operates in the maturation of neonatal mouse islets during ex vivo culture (Figure 4 and Figure 5), demonstrating the conserved maturation mechanism in both zebrafish and rodents.

Also, the zebrafish islet in the paper is divided into a mantle and core, but with only 15-20 β-cells making up the islet, it is questionable if there truly is a mantle/core distinction such as the one that is readily apparent in rodent islets.

We appreciate your concern. A key benefit of our two-photon light-sheet microscope (2P3A-DSLM) is its nearly isotropic resolution in 3D, which outperforms two-photon point-scanning microscope in resolving tightly packed structures in axial direction. As shown in Figure 2—figure supplement 1A, B, we could determine whether a cell was inside or at the periphery of the islet based on information obtained from 2D sections and 3D reconstructions. Core cells (enclosed by red lines) were classified as cells that were adjacent to other cells but not faced to the open field. We have added this information in our revised manuscript (Figure 2—figure supplement 1A and B and in the first paragraph of the subsection “Sequential initiation of β-cell functionality from the islet mantle to the core is coordinated by islet vascularization”). If we performed similar experiments using two-photon microscopy, its poor axial resolution would yield obscure 3D reconstructions that would prevent a clear-cut distinction of mantle and core cells.

The visualization of the vascularization of the islet during development is nice but would not have required a two-photon light-sheet setup.

As islets were buried deep within the zebrafish embryos, microscopes based on one-photon excitation, such as confocal or one-photon light-sheet microscope cannot resolve the boundary of individual β-cells. As compared to the point-scanning two-photon microscope, our two-photon light-sheet setup confers two obvious advantages:

1) As shown in Figure 2—figure supplement 1,we were able to clearly differentiate β-cells in the mantle or in the core, which could not be achieved with the two-photon point scanning microscope (see also responses above). The high spatial resolution also helped us to sort out the spatial relationship of islet vessels with glucose-responsive β-cells (Figure 2—figure supplement 2C-E).

2) Besides difference in axial resolution, our two-photon light-sheet microscope confers much reduced photo-bleaching as compared to the point scanning two-photon microscope (Zong, Zhao et al., 2015). We believe that low photo-bleaching and photo-toxicity are essential for monitoring β-cells function in vivo without perturbing their development and function (Videos 2-3 and 5-6).

The major concern is that throughout the paper, β-cell maturation and function are conflated. This betrays a serious flaw in the premise of the work that pervades the entire paper. Transgenic zebrafish are used that express the Rcamp red calcium indicator in β-cells. These fish are exposed to glucose and an Rcamp response to glucose is interpreted to indicate functional maturity of the responding β-cells. Using this approach, the investigators observe calcium responses first on the outside of the islet followed by β-cells in the islet core and conclude from this that β-cells mature from mantle to core. The problem with this interpretation is that if β-cells at the core were mature but were not exposed to glucose, they would not respond with a calcium response. Indeed, the investigators then demonstrate the vascularization proceeds from the mantle to the core and suggests that the mantle β-cells respond first simply because they are exposed to the circulation earlier in development. The experiments where vascularization is perturbed chemically or via genetic perturbation predictably cause reduced β-cell functionality, but likely not because β-cell maturation is inhibited, but because glucose exposure of functionally mature β-cells is limited due to the lack of circulation.

We thank you for pointing out this concern. To address this question, we have performed two new experiments.

1) Within 5 min after addition of 20 mM fluorescent deoxyglucose analog 2-NBDG to 56 hpf and 72 hpf cloche mutants, we did observe uptake of 2-NBDG into all of the Rcamp1.07-labeled β-cells in vivo even though blood circulation in these fish was completely absent (Figure 2—figure supplement 3A). In contrast, addition of 8 mM 2-NBDG failed to label the core β-cells of 56 hpf wild type fish (Figure 2—figure supplement 3B).

2) Additionally, by supplementing CGA (calcineurin activator) to cloche mutants, we were able to increase both the number and the function of glucose-responsive β-cells in the islet core at 72 hpf (Author response image 1).

Based on these data, we conclude that high glucose (20 mM) did penetrate into the islet core and reach all the β-cells within the islet even without circulation. The defective function of β-cells in the islet core of cloche mutants can be rescued by direct activation of calcineurin, again indicating that these cells can be accessed by high glucose. Therefore, islet microcirculation is essential for chronically delivering physiological dose of glucose to β-cells in the islet core for their functional development in vivo, but does not prevent their function to be evaluated under acute stimulation with supraphysiological dose of glucose. We have put all these supporting evidence in our revised manuscript (Figure 2—figure supplement 3, subsection “Sequential initiation of β-cell functionality from the islet mantle to the core is coordinated by islet vascularization”, third paragraph, subsection “Fine glucose concentrations regulate the heterogeneous development of β-cell function in vivo”, first paragraph and Figure 4, Figure 4—figure supplement 1C, subsection “Glucose-induced calcineurin/NFAT activation to initiate and enhance β-cell functionality”, first paragraph).

**Author response image 1. respfig1:** Supplementing CGA rescued the defective β-cell function in both the core and the mantle of the islet in 72 hpf *cloche* mutants. (**A-B**) In Tg *(ins:Rcamp1.07); cloche* mutants at 72 hpf, supplementing CGA significantly rescued the number (**A**) and the maximal Ca^2+^ influx (**B**) of glucose-responsive β-cells both in the islet core and in the islet mantle. **P* < 0.05, ***P* < 0.01; ns, not significant.

There is an attempt of demonstrating that mature β-cells exist in islets using a MafA antiserum, but there is no evidence that this antiserum works, let alone detects zebrafish MafA despite the considerably evolutionary distance between fishes and mammals (beyond the manufacturers claims it may, which is clearly insufficient validation). It is telling that the images of whole mount MafA staining appear to be cytoplasmic and do not overlap well with insulin, while MafA is a nuclear transcription factor of β-cells that typically requires antigen retrieval for detection.

Thank you for pointing out this issue. Per your suggestion, we have systematically evaluated this antibody and also other antibodies. In short, we conclude that none of the four commercially available antibodies (SAB2101414, Sigma; ABE1404, Millipore; A300-611A, Bethyl; IHC-00352, Bethyl) works in the zebrafish system. Therefore, we have completely deleted the related data from the manuscript. We have listed experiments that we have done in this regard below:

1) None of the current MafA antibodies labeled nucleus in zebrafish β-cells. We have tried to use the original MafA antibody (SAB2101414, Sigma) to label β-cells in isolated adult zebrafish islets. As shown below, they still displayed diffusive fluorescent signals in the cytosol, with no nuclear targeting signals. We also tested other available antibodies targeting possibly conserved epitopes of MafA (ABE1404, Millipore; A300-611A, Bethyl; and IHC-00352, Bethyl), whereas none of them labeled nucleus (data not shown).

**Author response image 2. respfig2:** MafA antibody used in mouse exhibited non-specific cytoplasmic signals in β-cells of zebrafish.

2) MafA expression was detected in the exocrine regions of pancreas rather than β-cells in zebrafish by in situ hybridization.Using in situ hybridization with anti-MafA probe, we could not detect any MafA expression in GFP-positive β-cells, but instead we detected MafA expression in the exocrine regions of pancreas in 12 dpf Tg *(ins: EGFP)* larvae (Author response image 3). This finding is consistent with the report that zebrafish MafA is not expressed in adult β-cells, but in exocrine pancreas (Matsuda, Mullapudi et al., 2017).

**Author response image 3. respfig3:** MafA was not expressed in the β-cells, but was expressed in the exocrine pancreas in larval zebrafish.

3) Expression of ucn3 in both α- and β-cells of zebrafish. Per another comment, we also tested whether ucn3 could be used to evaluate the maturity of β-cells in zebrafish embryos. To our surprise, we found that both β-cells and α-cells in zebrafish expressed high levels of ucn3 (Author response image 4), which is consistent with human islet data (van der Meulen, Xie et al., 2012) but arguing against the opinion that ucn3 is a specific maturation factor of β-cells.

**Author response image 4. respfig4:** ucn3 expression pattern in zebrafish islets at different developmental stages. (**A**) Immunofluorescent labelling of insulin and ucn3 in zebrafish islets at indicated stages. (**B**) Quantification of ucn3-positive β-cells in zebrafish islets at the indicated stages in A. n = 6-12 islets per condition. (**C**) Immunofluorescent labelling of glucagon and ucn3 in zebrafish islets at 72 hpf. (**D**) Quantification of ucn3-positive α-cells in zebrafish islets at 72 hpf. n = 10 embryos.

A similar problem arises for the experiments where calcineurin/NFAT is inhibited. Doing so inhibits calcium responses in vivo and is interpreted as a reduction in β-cell maturity. The possibility that blockade of calcineurin/NFAT blocks β-cell function is not considered, yet likely.

Per your suggestion, we have compared effects of acute and long-term incubation of calcineurin/NFAT inhibitors. As shown in Author response image 5, acute or short-term incubation of calcineurin/NFAT inhibitors (FK506 or VIVIT, less than 3 hrs) did not affect 20 mM glucose-stimulated calcium responses per se; in contrast, only after long-term incubation over 6 hrs did these β-cells exhibit reduced high glucose-stimulated calcium responses in vivo. This demonstrates that the calcineurin/NFAT inhibitors do not directly impair the β-cell functional machinery. The reduction in glucose-stimulated calcium responses was due to the diminished calcineurin/NFAT signaling pathway required for the β-cell maturation.

**Author response image 5. respfig5:** Blockade of calcineurin/NFAT led to reduced β-cell maturity. (**A-B**) In 60 hpf Tg (*ins:Rcamp1.07*) embryos, not acute/short-term, but only long-term incubation with calcineurin/NFAT inhibitors significantly reduced the number (**A**) and the maximal Ca^2+^ influx (**B**) of high glucose-responsive β-cells. After drug removal, the Ca^2+^ influx of some cells was recovered faster than the number of the glucose-responsive β-cells. **P* < 0.05, ***P* < 0.01.

After suggesting that maturation of β-cells starts at the mantle and progresses to the core, this model is transplanted to mouse β-cells. The rationale is that 'β-cells of neonatal mouse islets respond poorly to glucose and need more than one week to mature in vivo (Blum et al., 2012).' Actually, this paper by Blum showed that β-cells in the first days post-partum are more sensitive to glucose and that their glucose threshold increases as the islets mature. This was subsequently shown by others to be caused by the onset of intra islet feedback on β-cell function. It appears that the experiments on mouse islets in this current manuscript are based on only a handful of the pertinent papers that have either not been carefully read or have been mis-understood.

A key difference between Blum’s data and our previous data is that their neonatal β-cells released more insulin at low glucose condition than mature β-cells, which was not observed in our previous data. Thanks to your reminder, we have noticed that Blum et al. used low glucose (5.6 mM) to culture the neonatal islets. Therefore, we have systematically examined insulin secretion of neonatal and adult islets cultured under different concentrations of glucose. As compared to adult islets, neonatal islets cultured in medium with 5.6 mM or 7 mM glucose did exhibit more basal insulin secretion under 3 mM glucose stimulation(Figure 5A, B, D, E), which confirms Blum’s finding. Interestingly, low glucose-stimulated excessive insulin secretion of the neonatal islets was not affected by including CGA in the culture medium but was abolished by increasing the glucose concentration in the culture medium to 11/15/20 mM. These data have highlighted the profound impacts of ex vivo glucose concentration on neonatal β-cell functional development. Thus, to be able to image β-cell function in their native environment offers a clear advantage.

On the other hand, culturing neonatal islets with medium containing 5.6/7/11 mM glucose and CGA significantly enhanced high glucose-stimulated insulin secretion. In fact, with 11 mM glucose and CGA in the culture medium, GSIS of neonatal islets reached the maximum that is indistinguishable from the responses of adult islets. In this regard, β-cell maturation ex vivo composes of two processes: the reduction of low glucose-stimulated insulin secretion, and the increase of high glucose-stimulated insulin secretion. While both processes were enhanced by an optimal glucose concentration (11 mM) in the culture medium, calcineurin activator CGA selectively facilitated the latter process (Figure 5A, B, D, E).

Finally, we used mouse islets cultured in vitro to evaluate the correlation between glucose stimulated calcium transients and GSIS. As compared to adult islets, neonatal islets cultured in medium with 5.6 mM glucose also exhibited significantly higher ensemble calcium transients upon stimulation with 2.8 mM glucose; these 2.8 mM glucose-triggered calcium transients were abolished if neonatal islets were cultured in 11 mM glucose, and became indistinguishable from those of adult islets. Moreover, although including CGA in the culture medium did not reduce low glucose-triggered calcium transients in neonatal islets cultured in 5.6 mM glucose, it did enhance the maximal amplitude of calcium transients triggered by 16.7 mM glucose in neonatal islets cultured both in 5.6 and 11 mM glucose (Figure 5A, B, D, E). Therefore, glucose-induced calcium response is a reliable and sensitive marker of β-cell function status.

We have updated the related figures (Figure 5 and Figure 5—figure supplement 1) and incorporated related discussion in the manuscript (Discussion, second paragraph).

To demonstrate that maturation proceeds from mantle to core in mouse islets, the investigators assay glucose uptake using a fluorescent glucose tracer. They observe highest uptake in the outer layers of the islet and appear to interpret this as evidence in support for a model where maturation proceeds from the mantle to the core? But these islets are done in 8-week old islets, when the entire β-cell mass has long since matured. There is no evidence that the cells that take up glucose tracer at the periphery are actually β-cells, as the mouse islet mantle is the site of α- and δ-cells. I could not find any indication to rule out that the selective labeling of the mantle is caused by the limited diffusion of the tracer into the islet core.

We are sorry for the misunderstanding. This figure is not intended as evidence for the maturation model from the mantle to the core, but to show that low concentrations of glucose did not penetrate into the islet core as efficiently as they reach the islet mantle in the absence of blood circulation in isolated islets cultured in vitro. 2-NBDG is a fluorescent deoxyglucose analog that can be taken up by the cells through glucose transporters, thus its fluorescence can be used to quantify the glucose uptake by the cells. In contrast to a limited diffusion of 7 mM of the tracer into the islet core, we have tried to label islets from the transgenic mice whose α-cells are genetically labeled with EYFP with high concentration of 2-NBDG (20 mM). Under this condition, we were able to observe relatively homogenous labeling of cells within the islet core, in contrast to little labeling of α-cells at the islet periphery (Author response image 6). This result is consistent with the previously reported non-β-cells’ glucose uptake rate lower than that of β-cells (Heimberg, De Vos et al., 1995). To avoid the misunderstanding, we have used the following related zebrafish figures to replace this mouse data in the manuscript (Figure 2—figure supplement 3).

**Author response image 6. respfig6:** α-cells in isolated mouse islets almost did not take up 2-NBDG. Representative z-stack images of 2-NBDG (green) and EYFP (a red pseudo-color) signals in isolated 8-week-old mouse islets from the transgenic mice, whose α-cells are genetically labeled with EYFP after incubation with 20 mM 2-NBDG for 10 min.

Similarly, 8 mM 2-NBDG (glucose) did not reach the islet core in 56 hpf fish in the absence of islet circulation, but fully penetrated the whole islet at 72 hpf when islet circulation had developed (Figure 2—figure supplement 3B). In contrast, 20 mM 2-NBDG (glucose) readily penetrated the islet core in 56 hpf control fish, 56 hpf and 72 hpf cloche mutants that had no blood circulation (Figure 2—figure supplement 3A). Collectively, these data argue that physiological dose of glucose needs delicate intra-islet circulation to be delivered to the islet core, while supra-physiological dose of glucose does not require circulation to penetrate the islet core. Thus, β-cells in the core in vivo may exhibit arrested functional development because of inefficient endogenous glucose delivery without blood circulation, but their function could still be tested by acute application of supra-physiological dose of glucose.

In supporting this assertion, we have shown that CGA application rescued the dysfunctional β-cells in the islet core in cloche mutants (Figure 4), and also potently enhanced insulin secretion of neonatal mouse islets cultured in vitro (Figure 5). We have added this information in our revised manuscript (Figure 2—figure supplement 3, subsection “Sequential initiation of β-cell functionality from the islet mantle to the core is coordinated by islet vascularization”, third paragraph and subsection “Fine glucose concentrations regulate the heterogeneous development of β-cell function in vivo”, first paragraph). We also added the working principle of 2-NBDG in the Materials and methods in the new version of our manuscript (subsection **“**Uptake of the fluorescent D-glucose analogue by live zebrafish embryos”).

The effects of calcineurin on zebrafish β-cell function and maturation appear to confirm published work from over a decade ago. The novelty of these findings may be limited to the zebrafish pancreas field.

Instead of just confirming an effect of calcineurin on zebrafish β-cell function, we argue that our work is novel as listed below:

1) By developing an innovative microscopy method, we are able to directly visualize single β-cell function in a living organism in a non-invasive manner.

2) With this advanced technology, we have revealed the heterogeneous β-cell functional development in zebrafish in vivo, which highlighted an essential role of islet microcirculation that delivers optimal glucose to β-cells and activates downstream calcineurin/NFAT.

3) The current finding is not limited to zebrafish pancreas field. In mouse neonatal β-cells cultured in vitro, we have delineated two parallel processes contributing to the β-cell maturation. While optimal glucose is essential for the reduction of low glucose-stimulated insulin secretion, the combination of optimal glucose and calcineurin activator potentiates high glucose-stimulated insulin secretion to an extent indistinguishable from that of adult β-cells.

This finding may inspire stem cell researchers to use this previously unexplored strategy to promote the optimal maturity of the stem-cell derived β-like cells within clusters in vitro. Therefore, we believe our findings will attract common interest from both the top experts in the research field and the broader audience of *eLife*.

Additional comments from the reviewers:1) In several places in the text, the authors refer to β-cell proliferation. However, there are no assays actually performed to assess β-cell proliferation. Either the authors need to perform these experiments, or they need to remove such wording from the manuscript. They can refer to an increase in β-cell number, but in the absence of any direct evidence, one could not state with certainty whether this was from proliferation or neogenesis.

Thank you. Per your suggestion, we have made corrections in the new version of our manuscript.

2) The videos should be annotated to help the viewer. There were many times that I did not know what I was looking at, did not know when glucose had been added, if at all, and could not tell what the changes were that I should be focusing on.

Thank you for your suggestion. We have made new versions of viewer-friendly videos with annotations (Videos 3, 5 and 6in our revised manuscript). In addition, we have also added a new video to show the synchronized Ca^2+^ transients of β-cells within a live 72 hpf zebrafish embryo (Video 2).

3) In Figure 3C, the authors show that 20 mM glucose is not the optimal glucose concentration for the effects they are interested in. In fact, 8 mM glucose seems to be better. Why then do they use 20 mM glucose for most experiments?

Indeed, 8 mM glucose is the optimal concentration for long-term incubation to rescue β-cell function when endogenous glucose synthesis is suppressed by 3-MPA (an inhibitor of gluconeogenic phosphoenolpyruvate carboxykinase 1) during the late hatching period. In contrast, we transiently applied 20 mM glucose to acutely stimulate β-cells in vivo and in vitro to evaluate their function status.

4) In the Discussion, the authors refer to "locally synthesized glucose" as initiating β-cell maturation. What does this mean? What is the source of glucose? Along those lines, since feeding has not yet begun at 72 hpf, and the source of nutrition is the yolk, how do the authors correlate β-cell maturation in the fish with the changes noted in mice at the time of nursing initiation and switching to chow at weaning?

Prior to feeding, zebrafish embryos depend on passive diffusion or circulation of nutrition from yolk stores. The yolk syncytial layer (YSL) between the yolk and embryo proper is believed to regulate nutrient transfer from the yolk to the embryo proper. It starts to express gluconeogenic pck1 as early as 11 hpf, which is associated with increasing glucose in the embryos (Jurczyk, Roy et al., 2011). Thus, YSL is a potential site for endogenous glucose synthesis, and is in close proximity to the islet. We have added this information in our revised manuscript (subsection **“**Fine glucose concentrations regulate the heterogeneous development of β-cell function in vivo”, first paragraph). Besides, we also compared pancreas development between zebrafish and mouse in Supplementary file 1. Based on our findings and the literature, the period from 48 to 72 hpf in zebrafish equals to P2 and some days after P2 in mice. We speculate that 5 to 7 dpf in zebrafish when it begins to feed may correspond to the weaning stage in mice.

5) The authors have not "discovered" heterogeneity in β-cell maturation (Discussion, second paragraph). Their data do nicely support previous studies showing β-cell maturation heterogeneity looking at Ucn3 and MafA expression. Please cite these previous studies appropriately.

Thank you for the reminder. We have corrected the manuscript and cited the previous studies as suggested (Introduction, first paragraph).

6) The authors refer incorrectly to previous studies of islet vascularization in the mouse (Discussion, second paragraph). They are referred to Brissova et al. (2006, Diabetes) which shows very nicely that blood flow precedes islet morphogenesis. How do they interpret their findings in light of this previous study?

Thank you for raising this interesting question. We believe that our findings in fish are overall conserved to those of the mouse. Brissova et al. (2006) have demonstrated that islet blood flow establishes after the formation of small endocrine cell clusters, then “development of islet microvasculature and establishment of islet blood flow occur concomitantly with (or precedes) islet morphogenesis” (Brissova, Shostak et al., 2006). In our hands, although islet blood flow is absent in 48 hpf fish, it started to penetrate into the islet structure. Later this primitive islet structure will also undergo significant changes in the overall size and the position. In this sense, establishment of islet blood flow also occurs concomitantly with/precedes the final islet morphogenesis.

7) There are several data sets for which a t-test is not appropriate. Any time multiple treatments are being compared, an ANOVA should be used. Please consult with a statistician.

Thank you for the suggestion. In fact, all statistics were performed using GraphPad Prism 6 software (GraphPad Software Inc., La Jolla, CA). Unpaired Student’s two-tailed t-test was used to compare data between two indicated groups. We used one-way ANOVA followed by Dunnett’s test to do multiple comparisons with the control group. The description of statistical analysis has been corrected in the new version of our manuscript (subsection “Statistical analyses”).

8) In Figure 1—figure supplement 2, what is the rationale for the time points examined? Waiting 3 minutes until after glucose administration to start collecting the data seems too long. Likewise, taking an image once a minute seems too far apart. I am concerned that the authors missed something.

We actually collected data continuously both before and after glucose stimulation. As the oscillation frequency of calcium signals in pancreatic β-cells is in minute-scale (Grapengiesser, Dansk et al., 2003), we captured time-lapse images once a minute to prevent severe photo-bleaching in our previous experiment. To address your concern, we re-did the experiments at 1 Hz with 100 ms exposure time under a spinning-disc confocal microscope. We have replaced the previous figure with the new figure in our revised manuscript (Figure 1—figure supplement 2 and subsection “Visualization of embryonic β-cell function in vivo using 2P3A-DSLM”, first paragraph).

9) Are the VE+ cells the same cells that are Glu+ (Figure 2B). How do the authors define VE β-cells?

Our data showed that most of the Glu+ β-cells were VE+ β-cells, but not all the VE+ β-cells were Glu+ cells at different developmental stages (Figure 2—figure supplement 2C-E in our revised manuscript). We defined the VE+ β-cells as the β-cell located directly adjacent to vascular endothelia cells by using Tg (*ins:Rcamp1.07*); Tg (*flk1:GFP*) double transgenic zebrafish. We have put this information into supplementary materials in our revised manuscript (Figure 2—figure supplement 2C-E and subsection “Sequential initiation of β-cell functionality from the islet mantle to the core is coordinated by islet vascularization”, second paragraph).

10) In Figure 2C, what are the holes? α-cells, β-cells not expressing the reporter? Other endocrine cells? Why are there some cells outlines with no color assigned?

You are correct. The holes among β-cells are other kinds of endocrine cells, such as δ-cells and α-cells. Per your suggestion, we carefully checked again the raw data and our self-developed MATLAB program used for the inside-and-outside cell classification for the original Figure 2C. There was indeed a small bug in the source code of the program, which missed the cells with weak fluorescent signal. We have now corrected the Figure 2C as the new version of Figure 2A.

11) It needs to be made clearer that Figure 5A-C is from islets from 8 week old mice, not neonates.

Thank you. Per your suggestion, we have deleted the in vitro mouse data in previous Figure 5A-C, instead added the in vivo zebrafish data into the supplementary materials in our revised manuscript (Figure 2—figure supplement 3, subsection “Sequential initiation of β-cell functionality from the islet mantle to the core is coordinated by islet vascularization”, third paragraph and subsection “Fine glucose concentrations regulate the heterogeneous development of β-cell function in vivo”, first paragraph).

[Editors’ note: what now follows is the decision letter after the authors submitted for further consideration.]

Essential revisions:1) It is critical to address possible off target effects of the pharmacological reagents. A control experiment needs to be added showing that the calcium levels of a totally unrelated cell type (e.g. a CNS neuron) are not affected as observed in pancreas cells.

Following your suggestions, we have done additional control experiments to test possible off-target effects of these pharmacological reagents (10 mM BDM, 3 mM 3-MPA, 10 μm FK506 and 141.2 μm CGA) on neuronal activities. We measured calcium activities from CNS neurons in Tg *(elavl3:Gcamp6s)* zebrafish, in which neurons were labelled by the calcium indicator Gcamp6s. The result showed that the calcium dynamics in CNS neurons were not affected by any of these pharmacological treatments, supporting their specificity in affecting β-cell functional development only (Author response image 6). We have added this important information in our revised manuscript (Figure 4—figure supplement 2 and legend, subsection “Glucose-induced calcineurin/NFAT activation to initiate and enhance β-cell functionality”, last paragraph).

**Author response video 1. respvideo1:** 

2) The dominant negative experiments are not convincing. If the perturbing dominant negative construct does nothing more than make the cells thrive a little less, the obtained result (slowing the appearance of marker in the expressing cells) would be predicted to result. This is likely an artifact that needs to be carefully controlled.

Thank you for this insightful comment. We have sorted out two pieces of evidence that support the specific role of calcineurin/NFAT in β-cell functional development. Firstly, by adding numbers of insulin-positive cells at different stages to the histogram (Author response image 7, Figure 4E and legend, subsection “Glucose-induced calcineurin/NFAT activation to initiate and enhance β-cell functionality”, last paragraph), we show that genetic perturbation of calcineurin did not affect β-cell proliferation at different stages. In addition, we have examined the fish at another elder stage. Despite the progressive functional maturation of more β-cells in control embryos from 72 hpf to 7 dpf, the mean number and response of glucose-responsive β-cells in 7 dpf *dn-zCnA* embryos were not different from those in 72 hpf *dn-zCnA* ones. Therefore, β-cell functional maturation is specifically arrested at the stage of 72 hpf in *dn-zCnA* fish embryos.

**Author response image 7. respfig7:** Genetic perturbation of calcineurin/NFAT signaling persistently prevented the glucose-responsiveness of β-cells. (**A-B**) Numbers of glucose-responsive β-cells (**A**) and their maximal Ca^2+^ responses to glucose (**B**) in age-matched control and *dn-zCnA*-expressing embryos at 48 hpf, 72 hpf and 7 dpf. n = 5-7 embryos per stage. *p < 0.05, **p < 0.01, ***p < 0.001; ns, not significant.

3) The mouse studies are incomplete and seem to have been performed with inadequate attention to standardization. The Glucose Simulated Insulin Release (GSIS) studies are not presented in the text of the Results and in Figure 5. The calcium imaging and GSIS are performed at different concentrations.

Per your suggestion, we have standardized our results in mouse studies by adding two new graphs (Figure 5C and Figure 5F), showing the fold changes of GSIS (Figure 5C) and maximal amplitude of glucose-induced calcium influx under stimulation and resting conditions (Figure 5F). We also presented this information in the text of our revised manuscript (subsection “Direct activation of calcineurin promotes the high-glucose-stimulated secretion but does not reduce the low-glucose-stimulated secretion of neonatal β-cells in isolated mouse islets in vitro” and Figure 5 legend).

The calcium imaging and GSIS experiments were conducted at different glucose concentrations because: (1) GSIS assay in mouse was designed to be consistent with zebrafish studies, in which we used 20 mM glucose as the stimulation (Figure 5A-5C); (2) The calcium imaging experiments in mouse islets were added to address reviewer’s previous question, to show the similarities and differences of our work to the previous study from Douglas Melton group (Blum et al., 2012). Therefore, we used 2.8 mM and 16.7 mM glucose to follow Melton’s precedent. Nevertheless, we have done additional experiments to compare calcium activities of islets stimulated under different conditions (from 2.8 mM to 16.7 mM versus from 3 mM to 20 mM), which confirmed that these two conditions were not different. We do not think that it is necessary to include this identical result in the manuscript.

4) The finding should be more completely presented and fully addressed in the text and with appropriate statistical treatments.

Thank you for your kind reminder. Per your suggestion, we have added the quantitative information (statistical values with Mean ± SEM) throughout the Results section, and we have also updated some figures by adding new graphs with data standardization and more statistical analysis (Figure 1—figure supplement 4C, Figure 5C and Figure 5F). We believe that our findings are more completely presented and fully addressed in the text with appropriate statistical treatments in the revised manuscript.

Reviewer #1:[…] The authors address the potential role of glucose in the development of responsive β-cells, and offer several findings that are interesting, but seem incomplete. The studies showing the delivery of labeled glucose analogs show an apparent non-linearity that suggests there are unknowns not being addressed. A dose of a labeled glucose analog that is 2.5 times higher completely penetrates the core; the smaller dose completely fails, as described. The concentrations used in the glucose studies seem to range widely, from physiological to toxic.

Thank you and reviewer 3 for raising the question. We want to emphasize that all our glucose treatments can be sorted into two categories, “acute” and “chronic” treatments. Short-term application of 20 mM glucose is “acute” treatments that were used to evaluate the maximal responses of β-cells to glucose. Given the stimulation was discontinued and replaced with normal bath solution, it did not affect functions of islets cultured in vitro or in vivo in living fish embryos. On the other hand, long-term incubation of fish embryos or mouse neonatal islets with glucose (“chronic” treatments) was used to test potential role of glucose in functional acquisition during β-cell development. Under chronic circumstance, 20 mM glucose is toxic, which also agrees with numerous other reports on glucotoxicity (Wu et al., JBC, 2004; Robertson et al., Diabetes, 2004; Poitout et al., Biochim Biophys Acta, 2006).

The non-linear delivery of glucose into the islet core is an interesting phenomenon, which may be results of the balance between absorption of glucose via glucose transporters on cell membrane and diffusion of glucose in the gap between different cells. However, it is a topic out of the scope of the current study and will be a good direction for our future research.

The molecular studies on calcineurin and NFAT offer some interesting findings, that suggest involvement, but the results are presented in a fashion that makes it hard to view the issue as settled:"The first glucose responsive β-cells that appeared at 48 hpf were all EGFP negative, indicating that dn-zCnA prevents β-cells from acquiring glucose responsiveness."If the perturbation prevented the glucose responsiveness, it would not just be the first cells that are perturbed. Slowing could result from a variety of issues that are non-specific, just from expressing the perturbing transgene.

We are sorry for the misunderstanding and have removed that sentence from the manuscript. We have sorted out two pieces of evidence that support the specific role of calcineurin/NFAT in β-cell functional development. Firstly, by adding numbers of insulin-positive cells in different stages to the histogram (Author response image 7, Figure 4E and legend, subsection “Glucose-induced calcineurin/NFAT activation to initiate and enhance β-cell functionality”, last paragraph), we show that genetic perturbation of calcineurin did not affect β-cell proliferation at different stages. In addition, we have examined the fish at another elder stage. Despite the progressive functional maturation of more β-cells in control embryos from 72 hpf to 7 dpf, the mean number and response of glucose-responsive β-cells in 7 dpf *dn-zCnA* embryos were not different from those in 72 hpf *dn-zCnA* ones. Therefore, β-cell functional maturation is specifically arrested at the stage of 72 hpf in *dn-zCnA* fish embryos.

The mouse studies are incomplete and seem to have been performed with inadequate attention to standardization. The Glucose Simulated Insulin Release (GSIS) studies are not presented in the text of the Results and in Figure 5, the calcium imaging and GSIS are performed at different concentrations.

Per your suggestion, we have standardized our results in mouse studies by adding two new graphs (Figure 5C and Figure 5F), showing the fold changes of GSIS (Figure 5C) and maximal amplitude of glucose-induced calcium influx under stimulation and resting conditions (Figure 5F). We also presented this information in the text of our revised manuscript (subsection “Direct activation of calcineurin promotes the high-glucose-stimulated secretion but does not reduce the low-glucose-stimulated secretion of neonatal β-cells in isolated mouse islets in vitro” and Figure 5 legend).

The calcium imaging and GSIS experiments were conducted at different glucose concentrations because: (1) GSIS assay in mouse was designed to be consistent with zebrafish studies, in which we used 20 mM glucose as the stimulation (Figure 5A-5C); (2) The calcium imaging experiments in mouse islets were added to address reviewer’s previous question, to show the similarities and differences of our work to the previous study from Douglas Melton group (Blum et al., 2012). Therefore, we used 2.8 mM and 16.7 mM glucose to follow Melton’s precedent. Nevertheless, we have done additional experiments to compare calcium activities of islets stimulated under different conditions (from 2.8 mM to 16.7 mM versus from 3 mM to 20 mM), which confirmed that these two conditions were not different. We do not think that it is necessary to include this identical result in the manuscript. However, we could update Figure 5 in our revised manuscript if required.

This study has many interesting aspects to it, and it is clear that the approaches being deployed will offer important insights. It is within the authors' grasp to create a convincing and complete analysis of each of the findings they present, without any heroics or development of new technologies.The present version of the manuscript is a patchwork of incomplete studies that do not resolve any of the exciting aspects the study offers access to. I found the manuscript unconvincing, and incompletely presented. For example, it offers descriptions of results that are incompletely presented or misleading (see the comments on the penetration of labeled glucose analogs, and on the dn-zCnA studies).I would strongly recommend that the present version be rethought, and focus on presenting an analysis of either the zebrafish or the mouse, creating a substantial contribution to the literature.

We thank you for the suggestion on the tone of the paper and have toned down the tone in the paper accordingly. A few examples are listed here:

1) We have changed “we developed a high-resolution two-photon light-sheet microscope” into “we applied a high-resolution two-photon light-sheet microscope”.

2) We have deleted the part of calling calcineurin/NFAT signaling as the “master regulator” of β-cell functional development; instead we simply rephrased that calcineurin/NFAT is a major pathway for glucose-dependent β-cell functional development.

3) We have deleted this sentence “To our knowledge, this system is the first to achieve high-resolution in vivo imaging of individual β-cells and their functions.”

Regarding your and other reviewers’ comments on incompletely presented data, we have performed 3 new sets of control experiments and new standardizations of GSIS and maximal amplitude of glucose-induced calcium influx under stimulation and resting conditions (Figure 5C and Figure 5F). We have also added several quantitative information (statistical values with Mean ± SEM) throughout the Results section). We believe that the paper has been substantially improved and provides a complete, solid and interesting story.

Regarding the last comment of focusing only on either the zebrafish or the mouse, we want to set an example here: imaging the functional development of β-cells in vivo in the transparent zebrafish embryos provides mechanistic insights into this dynamic process, and the use of neonatal mouse islets to demonstrate the principles found in fish can be extended to mammalian species. In addition to all the findings demonstrated in the paper, we believe to establish such a precedent will also be a significant contribution to the field, which is best summarized by the reviewer 2 as “the microscopy approaches they developed and using the zebra fish islet model, is a great strategy to track the spatio-temporal islet developmental biology”.

Reviewer #3:The paper by Zhao et al. provides an in vivo analysis of developing pancreatic cells which is particularly challenging given that these cells are deep within the embryo.1) Their comparisons of different microscopes convincingly showed that their 2P3A-DSLM provided the best images. I see that they deconvolved their two-photon light-sheet data. Did they deconvolve the other data as well? One thing I could not determine from their methods was if they rotated the embryo to collect multiple views of each embryos to improve their Z-resolution (as in the classic SPIM paper)? This would also make their data more isometric. I think their spinning disc confocal data would have looked better with a higher magnification lens that the 10x they used.

Yes, images under 1P-SPIM, TPM and 2P3A-DSLM were all deconvolved with R-L algorithm by the Fiji software, as explained in Materials and methods in our manuscript. As the axial resolution of the 2P3A-DSLM is sufficient to resolve individual β-cells, we did not rotate embryos to enable fast sampling speed.

We agree with you that the quality of the spinning disc confocal data would be improved if an objective with higher magnification and high NA could be used. However, as the depth of the islets plus the thickness of the culture dish exceeds the working distance of the 25× or 40× objective lens, we were unable to achieve tight focus of the islet in living zebrafish embryos at magnifications greater than 10×.

2) Their finding that glucose responsive β-cells appeared in vivo earlier than previously reported (48 hpf instead of 52 hours) was very interesting.

We thank you for the appreciation of our finding.

3) The increase in calcium activity from the mantle to the core was another interesting finding. One question I had about the increase in calcium signaling from 56 to 72 hpf was whether these were the same cells or new cells with increased activity. Their data should easily reveal this, and it is an important point for them to mention.

Thank you for this insightful comment. Per your suggestion, we have plotted the histogram of maximal amplitudes of calcium transients from glucose-responsive β-cells in different developmental stages. From the right-shift of the histogram from 48 hpf to 72 hpf, it is obvious that the same population of β-cells gradually become more responsive to glucose. This is a good demonstration of the power of combining in vivo imaging with quantitative analysis. Thank you. We have added this information in our revised manuscript (Figure 1—figure supplement 4C and legend, subsection “Visualization of embryonic β-cell function in vivo using 2P3A-DSLM”, last paragraph).

4) Their hypothesis that blood vessels initiated β-cell functional development as measured by calcium activity may be true but the data from the cloche mutant and their pharmacological reagents (e.g. BDM) doesn't exclude another hypothesis, that β-cell functional maturation may cause these cells to secrete a factor that promotes angiogenesis. Remember blocking circulation early did not affect β-cells' ability to acquire glucose responsiveness. I am glad that the authors went beyond the correlation to actually test their hypothesis though it seems like their conclusions are still based more on correlation than causation.

*cloche^-/-^* mutant embryos and BDM-treated embryos have similar phenotypes: intact glucose-responsive β-cells at 56 hpf, but the same extent of impaired β-cells responsiveness to glucose at 72 hpf. These phenotypes support the conclusion that glucose, which is delivered to the β-cells through penetration from YSL at 56 hpf or through blood circulation at 72 hpf, is the trigger signal of β-cell functional development. On the other hand, you are perceptive in pointing out that we could not exclude the possibility that β-cell functional maturation may cause these cells to secrete a factor that promotes angiogenesis. When we used CGA to rescue their phenotypes, BDM-treated embryos showed more complete rescue of the maximal calcium influx than *cloche^-/-^* mutants (Figure 4—figure supplement 1), which may indicate possible involvement of vascular endothelial cells. However, more future experiments are needed to confirm or disprove the hypothesis.

5) I felt that the authors data from the cloche mutant and their dnCalcineurin line were more convincing than their data from pharmacological reagents for which I'm always concerned about off target effects. How specific are these particular reagents? The authors say they did careful dose response curves but the control I would really like to see is if the drugs affected calcium signaling in a totally unrelated cell like a CNS neuron. Do calcium levels change in these cells with drug treatment?

Following your suggestions, we have done additional control experiments to test possible off-target effects of these pharmacological reagents (10mM BDM, 3mM 3-MPA, 10uM FK506 and 141.2 μm CGA) on neuronal activities. We measured calcium activities from CNS neurons in Tg *(elavl3:Gcamp6s)* zebrafish, in which neurons were labelled by the calcium indicator Gcamp6s. The result showed that the calcium dynamics in CNS neurons were not affected by any of these pharmacological treatments, supporting their specificity in affecting β-cell functional development only (Figure 4—figure supplement 2 and Author response video 1). We have added this important information in our revised manuscript (Figure 4—figure supplement 2 and legend, subsection “Glucose-induced calcineurin/NFAT activation to initiate and enhance β-cell functionality”, last paragraph).

I would be more convinced by data from Calcineurin or NFAT mutants. I'm not fond of calling calcineurin/NFAT signaling a "master regulator" as they did in Figure 4. The authors state that calcineurin/NFAT signaling is a major pathway for glucose mediated β-cell development but that didn't seem to be the case for outer cells at 56 hpf?

Thank you, we have deleted the part of calling calcineurin/NFAT signaling as the “master regulator” of β-cell functional development; instead we simply rephrased that calcineurin/NFAT signaling is a major pathway for glucose-dependent β-cell functional development.

6) I'm a little surprised that the data in Figure 3 did not show larger differences. I do agree that the 20 mM Glucose concentration is likely toxic and so it shouldn't be used.

It is interesting that glucotoxicity in fish embryo caused by chronic incubation of 20 mM (Figure 3) is less than mouse islet cultured in vitro (Figure 5). A living organism may have some buffering mechanisms to antagonize the glucotoxicity of 20 mM glucose to β-cells. This subtle difference highlights the importance of performing high-resolution imaging in vivo.

7) For the mouse data the authors used 20 mM Glucose level to activate insulin. Is this concentration not toxic for mouse islet cells? I was also a little confused by the calcineurin/NFAT independence at low glucose level but dependence at high glucose levels. In the Discussion it was a little difficult to compare the mouse and fish data because insulin levels were only measured in the mouse. For mouse cells they had to use a drug since they didn't have the dnCalcineurin transgenic they used in fish.

Thank you and reviewer 1 for raising the question. We want to emphasize that all our glucose treatments can be sorted into two categories, “acute” and “chronic” treatments. Short-term application of 20 mM glucose is “acute” treatments that were used to evaluate the maximal responses of β-cells to glucose. Given the stimulation was discontinued and replaced with normal bath solution, it did not affect functions of islets cultured in vitro or in vivo in live fish embryos. On the other hand, long-term incubation of fish embryos or mouse neonatal islets with glucose (“chronic” treatments) was used to test potential role of glucose in functional acquisition during β-cell development. Under such circumstance, 20 mM glucose is toxic, which also agrees with numerous other reports on glucotoxicity in β-cells (Lan Wu et al., JBC, 2004; R. Paul Robertson et al., Diabetes, 2004; Poitout et al., Biochim Biophys Acta, 2006).

Our experiments on neonatal mouse islets agreed nicely with previous works (Blum et al., 2012), which demonstrated mouse β-cell maturation involves the reduction in low-glucose-stimulated insulin secretion. Furthermore, we showed that glucose-stimulated calcium imaging data is faithfully consistent with GSIS results in our ex vivo study with neonatal mouse islets (Figure 5). Therefore, despite that it is impossible to measure insulin secretion from individual zebrafish embryos, we believe that glucose-stimulated calcium responses is a good approximation for β-cell function in fish embryo in vivo. In this sense, although we only used drugs to manipulate the calcineurin/NFAT pathway in mouse islets cultured in vitro, we did reach conclusion consistent with zebrafish experiments regarding the β-cell functional development: calcineurin/NFAT mediates the inductive roles of glucose in increasing high-glucose-stimulated insulin secretion.

[Editors' note: further revisions were requested prior to acceptance, as described below.]

Required changes:1. The response to reviewer #3, point 4 is great but it's not in the paper. Please include in the manuscript a sentence or two from the response to this reviewer. Same for reviewer #3, point 6. Why not mention that a living organism may have some buffering mechanism in the relevant part of the paper? That would be a nice Discussion point.

Thank you very much. Following your suggestions, we have included both information corresponding to reviewer #3 point 4 and point 6 in the text of our revised manuscript with following short sentences:

“Nevertheless, we could not exclude the possibility that β-cell functional maturation may cause these cells to secrete factors that promote angiogenesis, or completely eliminate the possible involvement of vascular endothelial cells in β-cell functional development.” (Subsection “Sequential initiation of β-cell functionality from the isletmantle to the core is coordinated by islet vascularization”, last paragraph.)

“However, it is interesting that glucotoxicity in fish embryo caused by chronic incubation of 20 mM is less severe than expected, suggesting that living organism may have some buffering mechanisms to antagonize the glucotoxicity of 20 mM glucose to β-cells.” (Subsection “Fine glucose concentrations regulate the heterogeneous development of β-cell function in vivo”, last paragraph).

2. In the figure legends, it might be good to mention that inside = core and outside = mantle. In the text they only use the terms core and mantle which is appropriate.

Thank you. Following your suggestion, we have mentioned this information “inside = core and outside = mantle” at its first appearance in the Figure 2 legend of our revised manuscript.

3. Subsection “Fine glucose concentrations regulate the heterogeneous development of β-cell function in vivo”, first paragraph: “periphery” should be “peripheral”.

Thank you. We have corrected the grammatical mistake in our revised manuscript.